# GPAS: Accelerating Convergence of LLM Pretraining via Gradient-Preserving Activation Scaling

**Tianhao Chen**[1*], **Xin Xu**[1*], **Zijing Liu**[2], **Pengxiang Li**[3], **Xinyuan Song**[4],
**Ajay Kumar Jaiswal**[5], **Fan Zhang**[1], **Jishan Hu**[1], **Yang Wang**[1], **Hao Chen**[1],
**Shizhe Diao**[6], **Shiwei Liu**[7], **Yu Li**[2], **Lu Yin**[8†], **Can Yang**[1†]

[1]The Hong Kong University of Science and Technology    [2]International Digital Economy Academy
[3]Dalian University of Technology    [4]Emory University    [5]University of Texas at Austin
[6]NVIDIA    [7]University of Oxford    [8]University of Surrey
{tchenbb, xxuca}@connect.ust.hk, l.yin@surrey.ac.uk, macyang@ust.hk

## Abstract

Modern Large Language Models, such as the LLaMA, Qwen and DeepSeek series, predominantly adopt the Pre-LayerNorm (Pre-LN) Transformer architecture. While being stable during pretraining and scalable to large model sizes, Pre-LN suffers from an exponential growth in activation variance across layers, causing the shortcut to dominate over sub-layer outputs in the residual connection and limiting the learning capacity of deeper layers. To mitigate this issue, we propose **Gradient-Preserving Activation Scaling** (GPAS), a simple technique that can be used in combination with existing approaches. GPAS works by scaling down the intermediate activations while keeping their gradients unchanged. This leaves information in the activations intact, and avoids the gradient vanishing problem associated with gradient downscaling. Extensive experiments across various model sizes from 71M to 1B show that GPAS achieves consistent performance gains. Beyond enhancing Pre-LN Transformers, GPAS also shows promise in improving alternative architectures such as Sandwich-LN and DeepNorm, demonstrating its versatility and potential for improving training dynamics in a wide range of settings. Our code is available at https://github.com/dandingsky/GPAS.

## 1 Introduction

The pursuit of more cost-effective and performant architectures is a central theme in the development of Large Language Models (LLMs). The introduction of the Transformer architecture [1] laid a solid foundation for modern language modeling and has since become the backbone of most state-of-the-art LLMs. To facilitate stable and scalable training, recent models such as the LLaMA, Qwen and DeepSeek series [2, 3, 4] adopt the Pre-LayerNorm (Pre-LN) Transformer architecture [5, 6, 7], enabling training models with up to hundreds of billions of parameters.

Although Pre-LN Transformers offer strong scalability and stability, they remain suboptimal in terms of convergence speed and parameter efficiency. Prior studies [8, 9, 10] have shown that deeper layers of Pre-LN Transformers can often be pruned or removed entirely with minimal impact on performance. While this creates opportunities for reducing inference costs, it also highlights a significant inefficiency in the training process—where most of the computational budget is spent.

One of the core reasons for the underutilization of deep layers lies in the accumulation of activation variance in Pre-LN Transformers. While the original Transformer architecture places Layer

---

* Equal contribution.
† Corresponding author.

39th Conference on Neural Information Processing Systems (NeurIPS 2025).

Normalization [11] after the residual connection (commonly referred to as Post-LN), Pre-LN places it before the attention and FFN modules, leaving the shortcut branch unnormalized (Figure 1a). As a result, activation variance tends to accumulate with layer depth—often at an exponential rate. Recent works [12, 13] have shown that this variance growth causes the signal from attention and FFN modules to be increasingly overpowered by the shortcut branch, limiting the effectiveness of deeper layers in transforming hidden states.

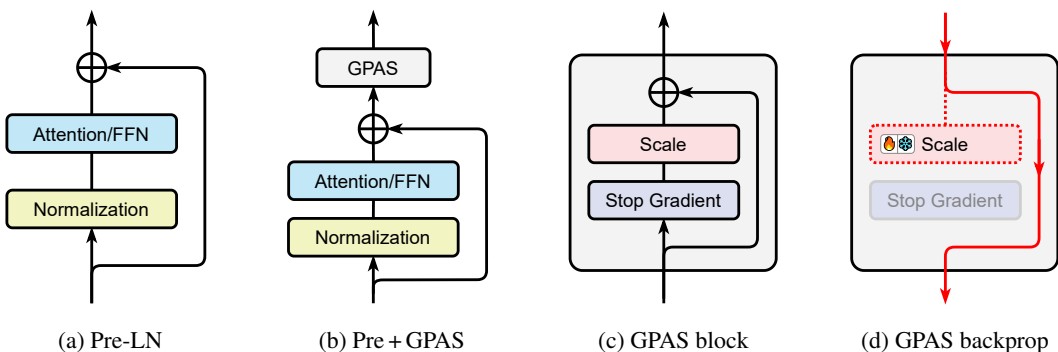

(a) Pre-LN        (b) Pre + GPAS        (c) GPAS block        (d) GPAS backprop

Figure 1: Architectures of Pre-LN, Pre + GPAS, and the GPAS block. Red lines indicate gradient flow. Red dotted line means the scale parameter can be either learnable or fixed.

In this work, we propose **Gradient-Preserving Activation Scaling** (GPAS), a simple yet effective solution to mitigate variance growth in Pre-LN Transformers. The idea of GPAS is straightforward: reduce variance growth by directly scaling down intermediate activations. However, naively scaling down activations in the forward pass leads to downscaled gradients during backpropagation, which could lead to vanishing gradients and slow down the learning process. As shown in Figure 1, GPAS works around this issue by scaling down forward activations while preserving the magnitude of backward gradients. Experiments on models ranging from 71M to 1B parameters show that Pre-LN with GPAS achieves faster convergence and improved downstream performance. Detailed analysis reveals that GPAS-augmented models exhibit a more compact distribution of activation variance across layers and more uniform layerwise importance.

Beyond enhancing Pre-LN Transformers, GPAS also serves as a general-purpose plugin for other architectural variants. We apply GPAS to architectures including DeepNorm [14], Sandwich-LN [15], Mix-LN [12] and LayerNorm Scaling [13], and observe consistent improvements in convergence speed and performance.

Our main contributions are summarized as follows:

- We propose Gradient-Preserving Activation Scaling to alleviate activation variance growth in Pre-LN Transformers while preserving gradient magnitudes.
- We validate the effectiveness of GPAS through pretraining experiments across various model sizes, and show that its benefits carry over to downstream supervised finetuning tasks.
- We further apply GPAS to other architectures such as DeepNorm, Sandwich-LN, Mix-LN and LayerNorm Scaling, and observe consistent performance gains.
- We conduct a thorough analysis of the training dynamics and model properties of models with and without GPAS.

## 2 Related Work

**Normalization layers.**    Normalization is crucial for training deep neural networks, especially for modern LLMs with increasing depth and parameter count. Layer Normalization [11] plays a vital role in the success of the original Transformer [1]. Despite its effectiveness, improving normalization remains an active area of research. GroupNorm [16] divides channels into subgroups and normalizes within each group. RMSNorm [17], which omits mean centering and uses root mean square instead of variance, offers a simpler and more efficient alternative to LayerNorm, and has gained popularity in recent LLMs [2, 3, 4]. AdaNorm [18] proposed a new transformation function to replace the gain

and bias in LayerNorm. Dynamic Tanh [19] eliminates normalization altogether by replacing it with a non-linear activation.

**LLM normalization schemes.**    Modern LLM architectures can be broadly characterized by three key components: normalization schemes, attention mechanisms, and FFN designs. Among these, normalization is most relevant to our work, while attention and FFN variants are largely orthogonal.

Transformer normalization has evolved around two main paradigms—Post-LN and Pre-LN, which have inspired numerous refinements to address their respective limitations. Post-LN was proposed in the vanilla Transformer [1], while Pre-LN was proposed in several works [5, 6, 7] to address the optimization issue of Post-LN. A theoretical perspective from [20] showed that Post-LN leads to larger gradients near the output layer, necessitating warm-up to avoid divergence. Conversely, Pre-LN yields smaller gradients in deeper layers, mitigating instability at initialization but also potentially curtailing the effectiveness of late-stage parameters. B2T [21] bypasses normalization in early layers to combat gradient decay of Post-LN. DeepNorm [14] modifies Post-LN by scaling up the residual connection before normalization and downscaling layer parameters during initialization. Sandwich-LN [15, 22] applies normalization both before and after the attention and FFN modules to further stabilize Pre-LN. Mix-LN [12] combines Post-LN and Pre-LN layers in the same model to alleviate both the variance growth of Pre-LN and the gradient vanishing issue of Post-LN. LayerNorm Scaling [13] scales output of normalization layers by inverse square root of layer depth, which facilitates a linear variance growth in Pre-LN.

As for attention and FFN architectures, existing variants mainly focus on expressiveness, scalability and hardware efficiency [23, 24, 25, 26]. While these directions are orthogonal to our work, we briefly mention them here to provide a more complete picture.

## 3   Gradient-Preserving Activation Scaling

### 3.1   Pre-LN with GPAS

We first demonstrate how to incorporate GPAS into the prevailing Pre-LN architecture, before extending to other baselines. Equation (1) shows a typical forward pass of a Pre-LN sub-layer (of layer $l$). GPAS introduces a simple modification after each sub-layer as Equation (2). Each layer now has an additional learnable scalar $\alpha_l \in \mathbb{R}$. SiLU$(\cdot)$ is the Sigmoid Linear Unit [27]. sg$(\cdot)$ is the stop gradient operator, which acts as an identity mapping during forward pass, but gives zero gradient during backpropagation. A visualization of this modification is shown in Figure 1.

The learnable gate $\alpha_l$ controls the amount of scaling that is applied to the intermediate activation $x'_{l+1}$. Moreover, by scaling $x'_{l+1}$, $\alpha_l$ controls the mixing ratio between shortcut $x_{l+1}$ and the output of the next sub-layer $f(\text{LN}(x_{l+1}))$, effectively balancing information from these 2 variables. SiLU activation is used to encourage positive $\alpha_l$ values to reduce activation variance, while also allowing the network to learn negative values when necessary. Notice that the input to the next sub-layer's attention or FFN is not scaled regardless of the value of $\alpha_l$. This is because Pre-LN architecture applies LayerNorm right before the attention or FFN module, which cancels out any scaling applied to $x'_{l+1}$. Overall, Equation (2) scales activation by $(1 - \text{SiLU}(\alpha))$, while the Jacobian $\partial x_{l+1}/\partial x'_{l+1} = I$.

$$\text{Pre-LN:} \quad x'_{l+1} = x_l + f(\text{LN}(x_l)), \quad f \in \{\text{Attn}, \text{FFN}\}, \tag{1}$$

$$+\,\text{GPAS:} \quad x_{l+1} = x'_{l+1} - \text{SiLU}(\alpha_l) \cdot \text{sg}(x'_{l+1}). \tag{2}$$

**Preserving gradient with stop gradient.**    We analyze the gradient preservation of GPAS using the first layer's gradient as example. For brevity only one sub-layer per layer is considered. Let $\beta_l = 1 - \text{SiLU}(\alpha_l)$ be the scaling factor for each layer. If we replace sg$(\cdot)$ with identity, Equation (2) becomes naive scaling: $x_{l+1} = x'_{l+1} \cdot \beta_l$. Suppose we have $L$ transformer layers in total, with $\mathcal{L} = \mathcal{L}(x_L, y)$ being the loss. Then the gradient $\partial \mathcal{L}/\partial x_1$ is:

$$\text{naive scaling:} \quad \frac{\partial \mathcal{L}}{\partial x_1} = \frac{\partial \mathcal{L}}{\partial x_L} \prod_{l=1}^{L-1} \frac{\partial x'_{l+1}}{\partial x_l} \cdot \prod_{l=1}^{L-1} \beta_l,$$

$$\text{GPAS:} \quad \frac{\partial \mathcal{L}}{\partial x_1} = \frac{\partial \mathcal{L}}{\partial x_L} \prod_{l=1}^{L-1} \frac{\partial x'_{l+1}}{\partial x_l}.$$

Since $\beta_l < 1$, the product $\prod_{l=1}^{L-1} \beta_l$ decays exponentially with depth, causing gradient vanishing problem in early layers. Applying GPAS eliminates the scaling terms $\beta_l$ during backpropagation and preserves gradient magnitude. Detailed derivations are provided in Appendix C.

## 3.2 Apply GPAS to Other Normalization Schemes

LayerNorm Scaling (LNS) [13] scales normalized output by inverse square root of layer depth with Equation (3). Since the scaling factor can be absorbed into the affine transformation inside LayerNorm, LNS is essentially Pre-LN with a different initialization of LayerNorm weights. Thus, we apply GPAS to LNS in the same way as Pre + GPAS.

$$\text{LNS:} \quad x'_{l+1} = x_l + f(\text{LN}(x_l)/\sqrt{l}), \quad f \in \{\text{Attn}, \text{FFN}\}. \tag{3}$$

$$\text{Sandwich-LN:} \quad x'_{l+1} = x_l + \text{LN}(f(\text{LN}(x_l))), \quad f \in \{\text{Attn}, \text{FFN}\}. \tag{4}$$

For Sandwich-LN [15], GPAS also applies after the residual connection with Equation (2). The only difference from Pre-LN is that the sub-layer forward has an additional LayerNorm as shown in Equation (4).

For Post-LN and DeepNorm [1, 14], however, the scenarios are quite different. As shown in Equation (5), Post-LN breaks the residual connection by wrapping $x_l + f(x_l)$ with LayerNorm. DeepNorm inherits this architecture and introduces a scaling factor $\alpha$ to the shortcut $x_l$ and another scaling factor $\beta$ to the sub-layer weights in $f$, where both $\alpha$ and $\beta$ are predefined and fixed during training. Empirically, we found applying GPAS after LayerNorm does not bring performance gain. Instead, we hypothesize that the scaling factor $\alpha$ is not optimal since it does not account for layerwise variation and training dynamics. Therefore, we apply GPAS to the scaled shortcut $\alpha \cdot x_l$, while keeping the input to attention and FFN unscaled. Ultimately, we found applying GPAS as Equation (7) and (8) boosts pretrain performance.

$$\text{Post-LN:} \quad x_{l+1} = \text{LN}(x_l + f(x_l)), \quad f \in \{\text{Attn}, \text{FFN}\}. \tag{5}$$

$$\text{DeepNorm:} \quad x_{l+1} = \text{LN}(\alpha \cdot x_l + f_\beta(x_l)), \quad f_\beta \in \{\text{Attn}, \text{FFN}\}. \tag{6}$$

$$\text{DeepNorm + GPAS:} \quad x'_l = x_l - \text{SiLU}(\alpha_l) \cdot \text{sg}(x_l), \tag{7}$$

$$x_{l+1} = \text{LN}(\alpha \cdot x'_l + f_\beta(x_l)), \quad f_\beta \in \{\text{Attn}, \text{FFN}\}. \tag{8}$$

Mix-LN [12] combines Pre-LN and Post-LN layers in the same model, so we apply GPAS differently to its Pre- and Post-LN layers. For the Post-LN layers, we apply the same strategy as Equation (7). For the Pre-LN layers, we use Equation (2).

# 4 Experiments and Results

## 4.1 Experiment Setup

**Model architectures.** Based on our proposed architectural modifications in Section 3, we conduct extensive experiments to verify their effectiveness. Following [28] and [29], we employ LLaMA-based model architectures for implementing Post-LN [1], DeepNorm [14], Pre-LN [30], Sandwich-LN [31], Mix-LN [12] and LNS [13]. All models share the same attention and FFN architectures as well as normalization layers except for normalization scheme. Specifically, all baseline architectures (from Post-LN to LNS) utilize RMSNorm [17], LLaMA attention and LLaMA MLP [32] with SwiGLU activation [33]. Moreover, they share the same initialization except for DeepNorm and LNS, which add additional scaling to certain parameters. Scaled Embed [34] is applied to stabilize pretraining. We then apply GPAS to these baseline architectures according to Section 3, and refer to the modified architectures as *DeepNorm + GPAS*, *Pre + GPAS*, *Sandwich + GPAS*, *Mix + GPAS*, and *LNS + GPAS*, respectively. Notice that we did not conduct experiment on Post + GPAS, since we found DeepNorm to be a better starting point in a preliminary study, and utilized our compute budget on DeepNorm + GPAS instead.

**Pretraining.** We perform pretraining experiments across all architectures and at five model scales: 71M, 130M, 250M, 350M, and 1B. For the larger 7B configuration, we only pretrain Pre-LN and Pre + GPAS due to limited computational resources. Following [28], we adopt the Adam [35] optimizer and train on the C4 dataset [36, 37]. We tokenize the pretraining corpus with T5 tokenizer [36]

since it was trained on the C4 dataset. **For models with GPAS, we initialize all learnable gates** $\alpha_l = 0$. When $\alpha_l = 0$, GPAS does not scale the activations or alter the gradients, meaning the very first forward and backward passes during training are identical to those of the baseline model. We did not explore alternative initialization schemes for $\alpha_l$. We also use the same gate value for both attention and FFN sub-layers within the same layer.

All experiments are carried out on $4 \times$ NVIDIA H800 GPUs. Models below 350M can be trained in under 1 to 8 hours, while each 1B model took 2 to 3 days of training. More detailed configurations are listed in Table 1.

Table 1: Pretraining configurations for models of different sizes.

| Model Size | Learning Rate | Warmup Steps | Training Steps | Batch Size | Train Tokens | Eval Tokens |
|---|---|---|---|---|---|---|
| 71M | $1 \times 10^{-3}$ | 1K | 10K | 512 | 1B | 10M |
| 130M | $1 \times 10^{-3}$ | 2K | 20K | 512 | 2B | 10M |
| 250M | $1 \times 10^{-3}$ | 4K | 40K | 512 | 4B | 10M |
| 350M | $5 \times 10^{-4}$ | 6K | 60K | 512 | 6B | 10M |
| 1B | $5 \times 10^{-4}$ | 10K | 100K | 512 | 10B | 10M |

**Supervised finetuning.** To determine whether the improvements observed during pretraining persist in subsequent training stages, we perform SFT on the pretrained models with 1B parameters from Section 4.2, and then evaluate their performance on common reasoning benchmarks. Following [12], we finetune the models on the Commonsense170K dataset [38] and evaluate the models on seven downstream tasks. **The learnable gates $\alpha_l$ are frozen during SFT to avoid disturbing pretrained knowledge.** We use a learning rate of $3 \times 10^{-4}$ and train for 4 epochs using LISA [39]. As for evaluation, we adopt the widely used LM Evaluation Harness [40].

## 4.2 Pretrain Results

Table 2: Perplexity ($\downarrow$) of pretrained models on evaluation set. Numbers in parentheses show improvement over the base method.

| Method | 71M | 130M | 250M | 350M | 1B |
|---|---|---|---|---|---|
| Post-LN [1] | 33.80 | 26.50 | 1351.58 | 21.19 | 1406.66 |
| DeepNorm [14] | 35.49 | 26.78 | 22.20 | 21.76 | 1400.39 |
| DeepNorm + GPAS | 34.78 (-0.71) | 26.62 (-0.16) | 21.89 (-0.31) | 21.29 (-0.47) | 16.01 (-1384) |
| Pre-LN [20] | 33.98 | 26.61 | 21.54 | 20.71 | 16.53 |
| Pre + GPAS | 33.38 (-0.60) | 26.25 (-0.36) | 21.34 (-0.20) | 19.77 (-0.94) | 16.11 (-0.42) |
| Sandwich-LN [15] | 32.28 | 25.31 | 20.43 | 20.20 | 16.26 |
| Sandwich + GPAS | **31.44** (-0.84) | **24.86** (-0.45) | 20.38 (-0.05) | 19.45 (-0.75) | 15.85 (-0.41) |
| Mix-LN [12] | 33.88 | 26.29 | 21.52 | 20.73 | 15.87 |
| Mix + GPAS | 33.26 (-0.62) | 26.03 (-0.26) | 21.43 (-0.09) | 19.82 (-0.91) | 15.38 (-0.49) |
| LNS [13] | 34.58 | 25.91 | 20.59 | 20.35 | 15.61 |
| LNS + GPAS | 32.68 (-1.90) | 24.95 (-0.96) | **19.89** (-0.70) | **19.38** (-0.97) | **14.87** (-0.74) |

Table 2 summarizes the pretraining results across all model sizes and architectures. Results for 7B-parameter Pre-LN and Pre + GPAS are provided in Appendix B. We have the following observations: ❶ **Consistent Gains from GPAS.** For nearly all architectures and model sizes, applying GPAS leads to lower perplexities (numbers in parentheses), confirming that preserving gradient magnitudes while adjusting activation scales is beneficial. ❷ **Best Baselines at Different Scales.** At smaller scale (71M), Sandwich-LN is the strongest baseline (32.28), and adding GPAS further reduces perplexity to 31.44. At larger scale (1B), LNS already outperforms other baselines (15.61), but GPAS still offers a notable improvement (14.88). ❸ **Magnitude of Improvement.** The improvements (in parentheses) range from moderate ($\approx$ 0.3–0.7 drop in perplexity) to quite substantial (e.g., $-1.90$ for LNS at 71M). These consistent gains highlight that GPAS can be seamlessly integrated with various normalization schemes and model sizes, providing an effective and lightweight solution to boost convergence and performance in LLM pertaining.

For a detailed analysis on how GPAS enhances performance, please refer to Section 5.

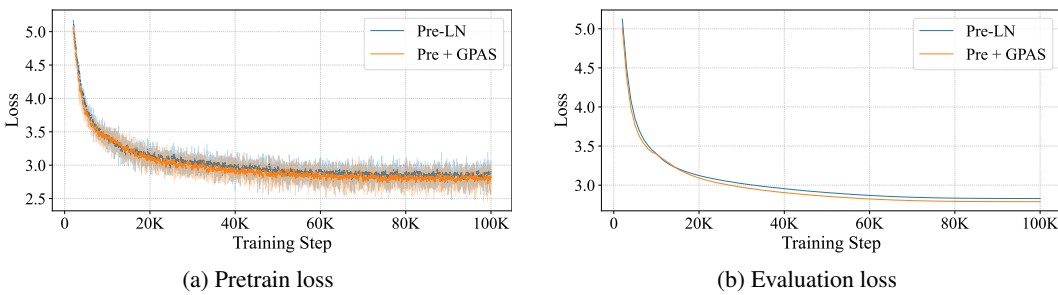

| (a) Pretrain loss | (b) Evaluation loss |

Figure 2: Pretrain and evaluation loss of Pre-LN and Pre + GPAS, 1B parameters

Table 3: 0-shot performance (↑) on various benchmarks (1B-parameter models). All numbers are accuracy in %. **Bold**: higher among GPAS and baseline method; ***Bold italic***: highest among all.

| Method | MMLU | BoolQ | PIQA | SIQA | HellaSwag | WinoG | ARC-e | ARC-c | OBQA | Average |
|---|---|---|---|---|---|---|---|---|---|---|
| Post-LN | 22.95 | 37.83 | 52.77 | 34.03 | 26.20 | 48.15 | 27.36 | 19.37 | 11.40 | 31.12 |
| DeepNorm | 22.95 | 37.83 | 52.77 | 34.08 | 26.20 | 51.14 | 27.31 | 19.37 | 11.40 | 31.45 |
| DeepNorm + GPAS | **26.46** | *62.11* | **69.53** | **46.93** | **34.37** | **52.09** | **49.24** | **22.61** | **20.40** | **42.64** |
| Pre-LN | 25.96 | 50.34 | 68.66 | 44.27 | 32.39 | 51.14 | 49.37 | 21.33 | 17.60 | 40.12 |
| Pre + GPAS | **26.68** | **59.79** | **69.31** | **46.52** | **33.64** | **52.49** | **49.79** | **22.70** | **22.00** | **42.55** |
| Sandwich-LN | **27.42** | 61.77 | 67.63 | 44.68 | 32.76 | **50.67** | 47.43 | 23.12 | 21.40 | 41.88 |
| Sandwich + GPAS | 27.29 | **61.90** | **69.15** | **45.29** | **34.61** | 50.36 | **51.39** | **23.46** | **22.20** | **42.85** |
| Mix-LN | **26.24** | 61.93 | 68.66 | 45.50 | 33.09 | 52.25 | 48.78 | **24.40** | 20.80 | 42.40 |
| Mix + GPAS | 26.23 | **61.99** | **69.59** | **45.60** | **33.51** | *53.51* | **50.34** | 22.35 | **22.40** | **42.83** |
| LNS | 26.62 | **62.02** | 69.48 | 45.39 | 34.76 | **51.38** | 50.88 | 23.29 | 19.80 | 42.63 |
| LNS + GPAS | *27.78* | 61.56 | *71.00* | *47.49* | *36.19* | 51.22 | *52.57* | *25.51* | *24.40* | *44.19* |

## 4.3 SFT Results

We present the finetuning results in Table 3 and notice the following observations: ❶ **Gains from GPAS Across Architectures.** When GPAS is applied (bottom half), every architecture sees a measurable boost in average accuracy. For example, Pre + GPAS achieved a notable 2.43% increase in average accuracy. DeepNorm + GPAS achieves the most dramatic improvement since vanilla Deep-Norm diverged during pretraining, while incorporating GPAS successfully avoided this catastrophic behavior. This suggests that DeepNorm's fixed residual scaling may not be optimal yet for network stability, and GPAS automatically learns to increase stability and convergence speed. ❷ **Task-Specific Improvements.** Breaking down the gains by individual datasets shows that the enhancements are typically consistent across tasks but vary in magnitude. For instance, Pre + GPAS's performance on BoolQ reached a 9.45% increase in accuracy, OBQA also has a notable gain of 4.4%. Meanwhile, LNS + GPAS attains improvements in both OBQA (+4.60) and HellaSwag (+1.43), pushing its overall average to 44.19% (+1.56).

Overall, by adaptively controlling the scale of the activations while preserving gradient magnitudes, GPAS consistently unlocks additional gains in downstream tasks.

## 5 Analysis

In this section, we provide a thorough analysis of GPAS by examining its training dynamics and model properties. We base our analysis primarily on Pre-LN and its GPAS-augmented version with 1B parameters. We also provide a detailed ablation study in Appendix A.

### 5.1 Learned Layerwise Gate Values

GPAS introduces a learnable gate $\alpha_l$ for each layer. We visualize the learned values of $\alpha_l$ in Figure 3. All models are the 1B-parameter models trained in Section 4.2. Generally, Pre-LN and Sandwich-LN layers tend to learn positive gates, while Post-LN layers tend to learn negative gates. Notably, for the Mix + GPAS model, the first 6 Post-LN layers learned negative gates, and the remaining Pre-LN layers learned positive values.

Across all model variants, the first layer tend to learn negative gate values. This means the network scales up the activation in the first layer. We hypothesize that this is due to the low variance of the initial word embeddings, and the network automatically learns to scale up that embedding to match the variance of subsequent layers. This hypothesis is also supported by Figure 4b, where the input activation to the first layer (which is the initial word embedding) has a much lower variance than the remaining layers.

Throughout training, the gates $\alpha_l$ exhibit the most variation during the first $10-20\%$ of steps. After that, they tend to remain relatively stable. One likely reason for this behavior is the use of half-precision training with BFloat16, which may suppress small updates to the gate values due to limited numerical precision.

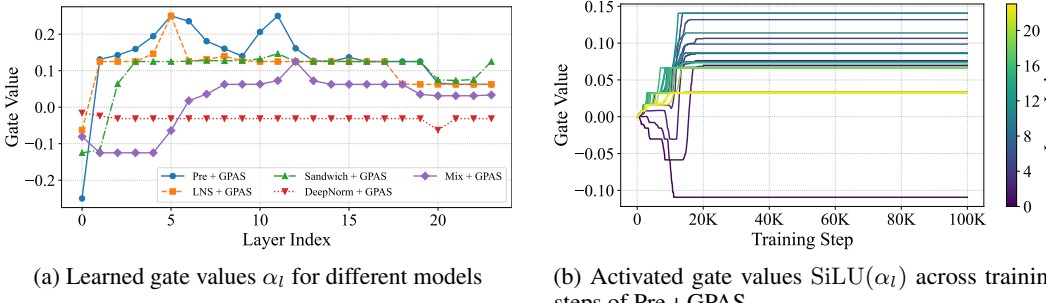

(a) Learned gate values $\alpha_l$ for different models   (b) Activated gate values $\mathrm{SiLU}(\alpha_l)$ across training steps of Pre + GPAS

Figure 3: Learned layerwise gate values

## 5.2   Activation Variance Comparison

We compare the activation variance of Pre-LN and Pre + GPAS, across all pretraining steps and layers. Experiments are conducted on 1B parameter models. We only plot every 2 layers including the first and last layers for brevity. Figure 4 shows that Pre-LN has an exponential increase in activation variance across layers, while Pre + GPAS offers a near 50% decrease in highest activation variance, with a more uniform and compact activation variance distribution across layers.

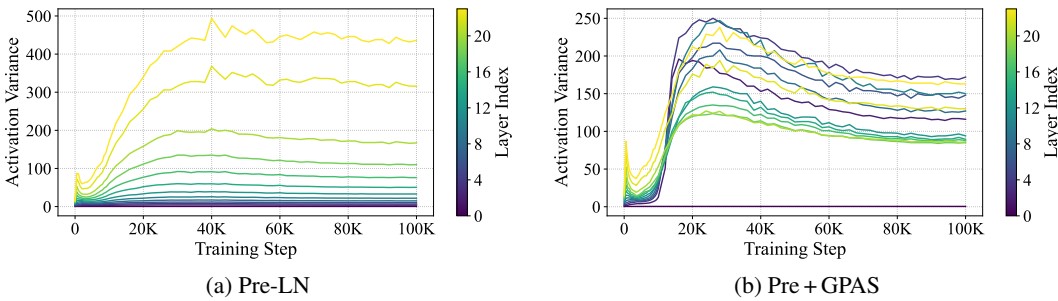

(a) Pre-LN   (b) Pre + GPAS

Figure 4: Layerwise activation variance with and without GPAS

## 5.3   Gradient Norm Comparison

We verify whether GPAS preserves gradient while scaling down activations. Figure 5 shows the layerwise gradient norms across across training steps. Models are the 1B parameter variants we trained in Section 4.2. It's evident that Pre + GPAS has a much larger gradient compared to its baseline method. Most layers in Pre-LN have gradient norm around $0.05-0.1$, while Pre + GPAS's are scaled to $0.05-0.5$. Although the first layer's gradient of Pre + GPAS seemed overly aggressive, the model was able to adjust to that gradient scale and achieve faster convergence.

However, we did find that the gradient spike around step 10K in Figure 5b disturbed the pretraining process. The gate values at step 10K went through an aggressive fluctuation (see Figure 3b), which caused the model to adjust to that change at steps $10K-30K$. This is also observed in the loss curves

in Figure 2, where both pretrain and evaluation loss of Pre + GPAS temporarily went above that of Pre-LN around step 10K. One simple way to fix this issue is to use gradient clipping on the gate values $\alpha_l$. However, we found that, while stablizing training, applying an additional gradient clipping to the gate values slightly degrades final performance (PPL 16.11→16.21). This suggests that the gate values might need an update scheme different from the main parameters. For our main experiments, we apply gradient clipping of 1.0 to all parameters except for the gates. For other GPAS-augmented baselines, we did not encounter such fluctuation in gate values which disturbs training.

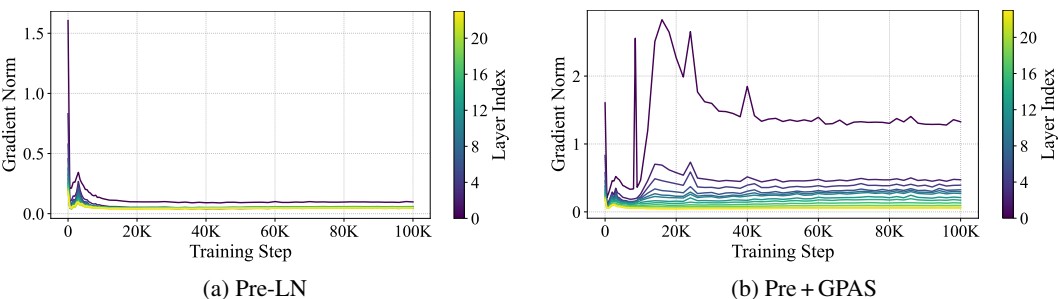

(a) Pre-LN

(b) Pre + GPAS

Figure 5: Layerwise gradient norm with and without GPAS

## 5.4 Weight Norm Comparison

Since Pre + GPAS scales up activations in the early layers according to Figure 4b, we suspect that its early layer weights also have larger norms. Specifically, we compare the norm of weights in the attention and FFN modules. Figure 6 shows that Pre + GPAS has larger weights in almost all layers compared to Pre-LN, especially in the early layers where activation variances are relatively small. This effectively scales up the output variance of attention and FFN modules, such that early layers and deeper layers share similar activation variance. We believe that the combination of larger weights and stable activation variance across layers enables the network to tolerate larger gradients without crashing, which in turn speeds up convergence.

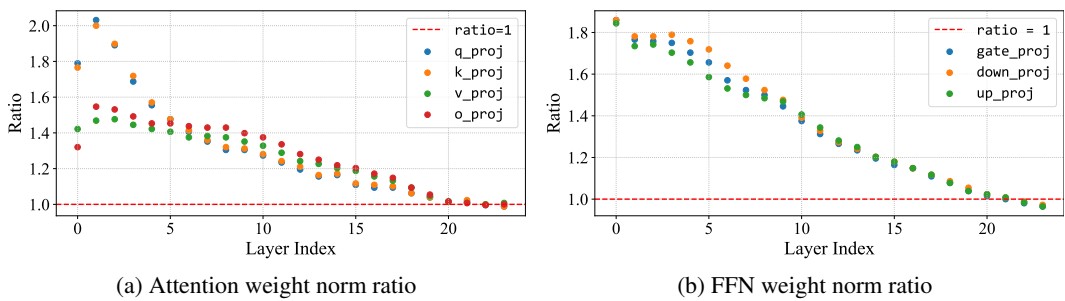

(a) Attention weight norm ratio

(b) FFN weight norm ratio

Figure 6: Attention and FFN weight norm ratios of Pre + GPAS over Pre-LN.

## 5.5 Layer Importance Comparison

We compare the layerwise importance of models with and without GPAS. The importance of a given layer $l$ is measured as the performance drop after removing that layer. We take the finetuned models in Section 4.3 as baseline, and measure the difference in average score across those benchmarks listed in Table 3 after removing each one of the layers.

Figure 7a shows that Pre + GPAS increases layer importance for most and especially the deeper layers, while some layers in vanilla Pre-LN are even harmful to performance. This indicates that GPAS enables a more efficient utilization of model parameters, while Pre-LN's exponential variance growth limited the learning capacity of deeper layers.

We also show layer importance for LNS + GPAS in Figure 7b. LNS mitigates variance growth in Pre-LN by downscaling LayerNorm output with square root of layer depth. In this case, GPAS still

amplifies the importance of each layer by a small but noticeable amount, leading to a higher overall performance.

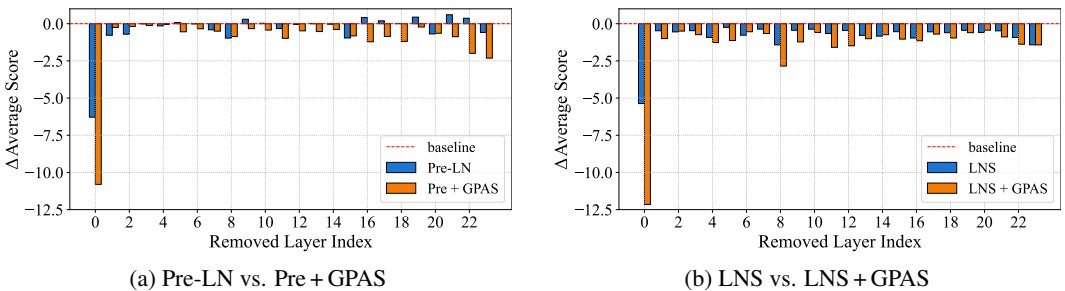

(a) Pre-LN vs. Pre + GPAS          (b) LNS vs. LNS + GPAS

Figure 7: Layer importance measured by performance drop after removal.

## 5.6 Gradient Analysis

Based on our experimental findings, we provide a brief and preliminary analysis to the activation variance and gradient profiles of Pre-LN and Pre + GPAS. By the result of [13] Theorem 1 and Lemma 1, and the work of [34], we have the following result for Pre-LN transformers:

**Lemma 1** (Variance and Gradient Growth in Pre-LN). *For a Pre-LN Transformer with $L$ layers, using Equations* (1)*, and let $W_\ell$ be the model parameter matrix at layer $\ell$. Assume that for all layers, $x_\ell$, $x'_\ell$, and $W_\ell$ are mutually independent and follow Gaussian distributions with mean zero. Let $y_L$ be the output of the $L$-th layer, and consider the gradient norm $\left\|\frac{\partial y_L}{\partial x_1}\right\|_2$. Let the upper bound for this norm denoted by $\mathrm{UP}(\cdot)$. Then it satisfies:*

$$M \leq \mathrm{UP}\left(\left\|\frac{\partial y_L}{\partial x_1}\right\|_2\right) \leq O(L), \tag{9}$$

When we add the GPAS to the Pre-LN transformers, we have the following theorem

**Theorem 1** (Variance and Gradient Growth in Pre-LN with GPAS). *Using Equations* (1) *and* (2)*, and under the same assumptions as in Lemma 1, assume that each $\alpha_\ell$ is bounded and varies slowly across layers. Let $L(\alpha)$ and $M(\alpha)$ denote the layerwise lower and upper bounds of $\log(1 - \mathrm{SiLU}(\alpha_\ell))$, respectively. Let $\sigma$ denote the variance of $x_\ell$. Then the upper bound of the gradient norm, denoted as $\mathrm{UP}\left(\left\|\frac{\partial y_L}{\partial x_1}\right\|_2\right)$, satisfies the following inequality:*

$$\exp\left\{O\left(\frac{1}{\sigma}\frac{1}{1 - e^{(-\frac{1}{\sigma+1} + L(\alpha))/2}} + 1\right)\right\} \leq \mathrm{UP}\left(\left\|\frac{\partial y_L}{\partial x_1}\right\|_2\right) \leq \exp\left\{O\left(\exp\left\{-\frac{1}{2}(M(\alpha))\right\}\frac{\min\{L, \sigma\}}{\sigma}\right)\right\} \tag{10}$$

The detailed description as well as the complete proof, are provided in Appendix D. From Theorem 1, By comparing Lemma 1 (before GPAS) with Theorem 1 (after GPAS), For the lower bound of the gradient norm upper estimate $\mathrm{UP}(\left\|\frac{\partial y_L}{\partial x_1}\right\|_2)$, the term $L(\alpha)$ effectively acts as a compensatory factor that offsets the influence of $\sigma$, thereby accelerating the growth of the bound beyond a constant rate. This leads to a strictly non-constant lower bound, which in turn ensures that the gradient magnitude retains sufficient variability across layers. In particular, this precludes the possibility of vanishing gradients and encourages meaningful gradient flow during backpropagation.

In contrast, for the upper bound, the term $M(\alpha)$ serves as a multiplicative scaling factor that exponentially suppresses the bound. This results in a significantly lower gradient norm ceiling compared to the $O(L)$ upper bound of standard Pre-LN Transformers. Consequently, the proposed modification mitigates gradient explosion and reduces the occurrence of large loss spikes, thereby enhancing training stability. Notably, the improved upper bound grows more slowly than $L$, further dampening excessive gradient amplification and promoting smoother optimization dynamics.

# 6 Conclusion

We proposed Gradient-Preserving Activation Scaling, a simple method that mitigates the exponential growth of activation variance in Pre-LN Transformers, thereby accelerating convergence and enhancing parameter efficiency. Beyond improving Pre-LN Transformers, GPAS also proves to be a viable plugin for alternative normalization schemes such as DeepNorm, Sandwich-LN, Mix-LN, and LNS.

**Limitations.**  While GPAS showed consistent gains in pretraining performance, our experiments are limited to 1B-parameter models due to computational constraints. Additionally, the current use of the SiLU activation may still permit unstable gate updates that disrupt training dynamics— although gradient clipping partially mitigates this issue. Nevertheless, the learnable gates introduce uncertainty during pretraining, and a predefined gate schedule with theoretical guarantees might be more preferable, especially at larger scales. A deeper understanding of how GPAS affects training stability and convergence remains an open question. Finally, GPAS is intended for training LLMs from scratch, and applying it to models pretrained without GPAS will likely be suboptimal.

**Broader Impact.**  GPAS offers a lightweight solution to improve the convergence speed and parameter efficiency of LLM pretraining. This may contribute to more accessible and sustainable development of foundation models, with potential downstream benefits across applications such as education and scientific research.

## Acknowledgments

This work was partially supported by a grant from the Research Grants Council of the Hong Kong Special Administrative Region, China (Project Reference Number: AoE/E-601/24-N).

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

# A    Ablation Study

We perform a series of ablation studies on GPAS to elucidate the rationale behind its architectural design. In particular, we investigate the impact of four key factors: the *Activation function*, the *Application location of GPAS*, the *Necessity of stop gradient operator*, and the choice between *Learnable vs. predefined gate value*. All ablations are conducted on 350M-parameter models, chosen because this scale typically serves as a reliable proxy for larger models while remaining feasible within our computational budget. We also limit the baseline architecture to Pre-LN only, and leave explorations of other architectures to future research.

**Activation function.**    GPAS uses the $\mathrm{SiLU}$ activation by default. To enable more customizable control over activated gate values, we generalize GPAS to Equation (11), where $\mathrm{Act}(\cdot)$ can be any activation function. We then examine how different activation functions affect performance, using a 350M-parameter Pre-LN model with GPAS. Table 4 shows that $\mathrm{Identity}$ and $\mathrm{Tanh}$ achieve slightly lower perplexities than $\mathrm{SiLU}$ in the 350M setting. However, this advantage does not persist at larger scale: in the 1B setting, $\mathrm{Identity}$ and $\mathrm{SiLU}$ achieve perplexities of 16.49 and 16.25, respectively. We hypothesize that these activations, particularly $\mathrm{Identity}$, permit overly aggressive gate updates during training; by contrast, $\mathrm{SiLU}$ imposes a smoother gradient profile, constraining updates to a more stable range.

$$x_{l+1} = x'_{l+1} - \mathrm{Act}(\alpha_l) \cdot \mathrm{sg}(x'_{l+1}). \tag{11}$$

Table 4: Ablations on activation function (left) and GPAS insertion position (right). Based on 350M Pre-LN. SiLU ($\beta = 8$) refers to scaled SiLU: $x \cdot \mathrm{sigmoid}(\beta x)$.

| Activation | Perplexity |
|---|---|
| Identity | 20.08 (**-1.27**) |
| ReLU | 21.17 (-0.18) |
| LeakyReLU | 21.17 (-0.18) |
| Tanh | 20.09 (-1.26) |
| SiLU ($\beta = 8$) | 20.26 (-1.09) |
| SiLU (default) | 20.35 (-1.00) |
| No GPAS | 21.35 |

| GPAS Position | Perplexity |
|---|---|
| After sub-layer (default) | 20.35 (**-1.00**) |
| Before sub-layer | 20.73 (-0.62) |
| After LayerNorm | 21.33 (-0.02) |
| After Attn / FFN | 21.23 (-0.12) |
| No GPAS | 21.35 |

**Where to apply GPAS.**    Since GPAS can, in principle, be integrated at any point where intermediate activations arise, we investigate the impact of inserting it at different locations within the transformer block. In Table 4, we compare four variants against a 350M-parameter Pre-LN baseline: (1) our default setting, which applies GPAS right after the sub-layer (residual + module output); (2) applying GPAS before the sub-layer; (3) placing it immediately after the LayerNorm; and (4) after each Attn/FFN module. While each insertion strategy outperforms the no-GPAS baseline, the default approach (inserting GPAS after the sub-layer) offers the largest perplexity reduction ($-1.00$), indicating that modulating the combined residual and module output is the most effective choice.

**Necessity of stop gradient operator.**    We conduct a control experiment to show the necessity of the stop gradient operator in GPAS. For the control experiment, we use Equation (12) to replace Equation (2):

$$x_{l+1} = x'_{l+1} - \mathrm{SiLU}(\alpha_l) \cdot x'_{l+1}. \tag{12}$$

We found that the stop gradient operator $\mathrm{sg}(\cdot)$ is crucial for preserving gradients. The gradient norm of GPAS without $\mathrm{sg}(\cdot)$ looks very similar to that of Pre-LN in Figure 5a. Although it still manages to scale down the activation variance by around 50% across all layers, GPAS without $\mathrm{sg}(\cdot)$ does not offer meaningful improvement in perplexity compared to Pre-LN (see Table 5). This highlights the importance of the stop gradient operator to prevent gradient vanishing issue associated with gradient downscaling.

**Learnable vs. predefined gate value.**    GPAS adopts learnable gates by default. We investigate whether predefined gates can also be used, which could potentially offer more predictability and consistency. To determine a reasonable set of predefined gate values, we extract the gates from

Table 5: Ablations on stop-gradient usage (left) and gating strategy (right) in GPAS.

| Method | Perplexity |
|---|---|
| w/ $\text{sg}(\cdot)$ | 20.35 (-1.00) |
| w/o $\text{sg}(\cdot)$ | 21.34 (-0.01) |
| No GPAS | 21.35 |

| Method | Perplexity |
|---|---|
| Learnable gate | 20.35 (-1.00) |
| Predefined gate | 22.46 (+1.11) |
| No GPAS | 21.35 |

pretrained Pre + GPAS model, and use those values to initialize the gates for the control experiment, where the gate values will be fixed during training. Results are shown in Table 5. We found that fixing the gates in this way led to a substantial drop in performance, especially in early stages when GPAS with learnable gates converges much quicker. This result confirms the importance of learnable gates to account for training dynamics. Another potential way of using predefined gate value is to introduce a warmup stage for the gates, which we leave to future work.

## B  Pretrain Results on 7B-Parameter Models

To further verify the effectiveness of GPAS on larger scale models, we perform pretraining experiments on Pre-LN and Pre + GPAS with 7B parameters. We follow [41] and use a learning rate of $3 \times 10^{-4}$ with 10K warmup steps and cosine decay. Batch size is set to 2048 and scheduled to train for 150K steps on 60B tokens. We use gradient clipping of $0.01$ on gate parameters $\alpha_l$ and $1.0$ on other parameters to stabilize training. Due to constraints on computational resources, we only train the models up to 40K steps with 16B tokens. After 40K training steps, Pre-LN and Pre + GPAS reached evaluation perplexity of 15.27 and 13.82, respectively. As shown in Figure 8b, Pre + GPAS achieves a much faster convergence than vanilla Pre-LN.

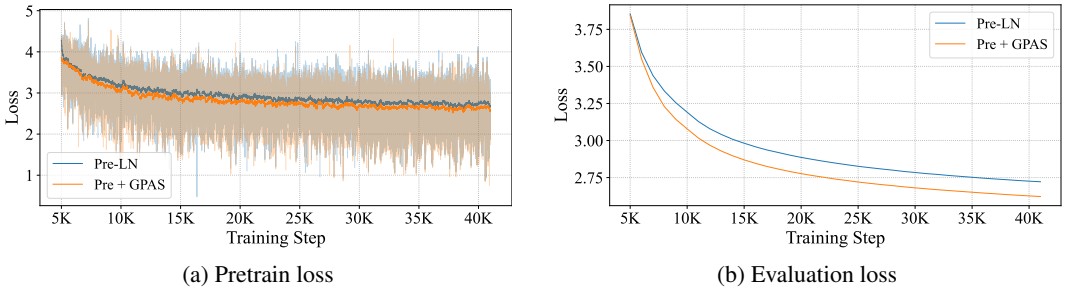

(a) Pretrain loss

(b) Evaluation loss

Figure 8: Pretrain and evaluation loss of Pre-LN and Pre + GPAS, 7B parameters

## C  Gradient Preservation with GPAS

Following Section 3.1, the gradient $\partial \mathcal{L}/\partial x_1$ is:

$$
\begin{aligned}
\text{naive scaling:}\quad \frac{\partial \mathcal{L}}{\partial x_1} &= \frac{\partial \mathcal{L}}{\partial x_L} \prod_{l=1}^{L-1} \frac{\partial x_{l+1}}{\partial x_l} \\
&= \frac{\partial \mathcal{L}}{\partial x_L} \prod_{l=1}^{L-1} \left( \frac{\partial x_{l+1}}{\partial x'_{l+1}} \frac{\partial x'_{l+1}}{\partial x_l} \right) \\
&= \frac{\partial \mathcal{L}}{\partial x_L} \prod_{l=1}^{L-1} \frac{\partial x'_{l+1}}{\partial x_l} \cdot \prod_{l=1}^{L-1} \frac{\partial x_{l+1}}{\partial x'_{l+1}} \\
&= \frac{\partial \mathcal{L}}{\partial x_L} \prod_{l=1}^{L-1} \frac{\partial x'_{l+1}}{\partial x_l} \cdot \prod_{l=1}^{L-1} \beta_l,
\end{aligned}
$$

$$
\begin{aligned}
\text{GPAS:}\quad \frac{\partial \mathcal{L}}{\partial x_1} &= \frac{\partial \mathcal{L}}{\partial x_L} \prod_{l=1}^{L-1} \frac{\partial x'_{l+1}}{\partial x_l} \cdot \prod_{l=1}^{L-1} \frac{\partial x_{l+1}}{\partial x'_{l+1}} \\
&= \frac{\partial \mathcal{L}}{\partial x_L} \prod_{l=1}^{L-1} \frac{\partial x'_{l+1}}{\partial x_l} \cdot \prod_{l=1}^{L-1} I \\
&= \frac{\partial \mathcal{L}}{\partial x_L} \prod_{l=1}^{L-1} \frac{\partial x'_{l+1}}{\partial x_l}.
\end{aligned}
$$

## D  Proof of Theorem 1 of GPAS

*Proof.* By equation (1) and 2, we have the following:

$$
\begin{aligned}
y = x_{\ell+1} &= x'_\ell + \mathrm{FFN}(\frac{1}{\sqrt{\ell}}\mathrm{LN}(x'_\ell)), \\
x'_\ell &= x_\ell + \mathrm{Attn}(\frac{1}{\sqrt{\ell}}\mathrm{LN}(x_\ell)).
\end{aligned}
\tag{13}
$$

$$
\text{Pre-LN: } x'_{l+1} = x_l + f(\mathrm{LN}(x_l)), \quad f \in \{\mathrm{Attn}, \mathrm{FFN}\} \tag{14}
$$
$$
\text{+ GPAS: } x_{l+1} = x'_{l+1} - \mathrm{SiLU}(\alpha_l) \cdot \mathrm{sg}(x'_{l+1}) \tag{15}
$$

Following the variance analysis in [13], both the FFN and Attn modules contribute equally to variance accumulation. For simplicity, we have:

$$
\sigma^2_{x_{\ell+1}} = \sigma^2_{x_\ell}(1 + \frac{1}{\sigma_{x_\ell}}) \cdot (1 - \mathrm{SiLU}(\alpha_l)). \tag{16}
$$

The variance with regard to $\sigma_{x_1}$:

$$
\sigma^2_{x_\ell} = \sigma^2_{x_1} \Theta\left( \prod_{k=1}^{\ell-1} \left(1 + \frac{1}{\sigma_{x_k}}\right)\left(1 - \mathrm{SiLU}(\alpha_l)\right) \right), \tag{17}
$$

For the parameter $\alpha_\ell$ we have

$$
1 - \mathrm{SiLU}(\alpha_\ell) = 1 - \frac{\alpha_\ell}{1 + e^{-\alpha_\ell}} = \frac{(1 + e^{-\alpha_\ell}) - \alpha_\ell}{1 + e^{-\alpha_\ell}}. \tag{18}
$$

$$
\log \sigma^2_{x_\ell} = \log \sigma^2_{x_1} + \sum_{k=1}^{\ell-1} \log\left(1 + \frac{1}{\sigma_{x_k}}\right) + \log\left(1 - \mathrm{SiLU}(\alpha_\ell)\right) + C, \tag{19}
$$

where $C$ is an (unspecified) constant. Using the original definition for SiLU we have

$$\log\Big(1 - \text{SiLU}(\alpha_\ell)\Big) = \log\Big(\frac{1 + e^{-\alpha_\ell} - \alpha_\ell}{1 + e^{-\alpha_\ell}}\Big) = \log\big(1 + e^{-\alpha_\ell} - \alpha_\ell\big) - \log\big(1 + e^{-\alpha_\ell}\big). \quad (20)$$

Next, denote $L(\alpha_\ell) = \log\Big(\frac{1 + e^{-\alpha_\ell} - \alpha_\ell}{1 + e^{-\alpha_\ell}}\Big)$, which is a function of $\alpha_\ell$. Thus the inequality becomes

$$\log\sigma_{x_\ell}^2 \geq \log\sigma_{x_1}^2 + \sum_{k=1}^{\ell-1}\Big(\frac{1}{\sigma_{x_k} + 1} + L(\alpha_\ell)\Big) + C. \quad (21)$$

For fixed $\sigma_{x_\ell}^2$, $\log\sigma_{x_\ell}^2 \geq L \cdot (C + L(\alpha_\ell))$, if $\alpha_l < 0$, there is no lower bound for the variance. So it can reach 0. To establish a upper bound for $\sigma_{x_\ell}^2$

$$\sum_{k=1}^{\ell-1}\log\Big(1 + \frac{1}{\sigma_{x_k}}\Big) \leq \sum_{k=1}^{\ell-1}\frac{1}{\sigma_{x_k}}. \quad (22)$$

The SiLU value of this term is less than or equal to zero (since typically $1 - \text{SiLU}(\alpha_\ell) \leq 1$) or it can be bounded by an appropriate constant $M(\alpha_\ell)$. For our derivation, we denote

$$\log\Big(1 - \text{SiLU}(\alpha_\ell)\Big) \leq M(\alpha_\ell). \quad (23)$$

Putting these bounds together we have

$$\log\sigma_{x_\ell}^2 \leq \log\sigma_{x_1}^2 + \sum_{k=1}^{\ell-1}\frac{1}{\sigma_{x_k}} + M(\alpha_\ell) + C. \quad (24)$$

Next, we analyze the gradient stability of GPAS. Following Equation (38) in [13], and applying the formalization techniques from [42], we derive the result under the consideration that $\mathrm{stopgrad}$ is used, thereby eliminating any gradient contributions at that point. Consequently, we obtain:

$$UP\Big(\Big\|\frac{\partial y_L}{\partial x_1}\Big\|_2\Big) = \prod_{l=1}^{L-1}\Big(1 + \frac{1}{\sigma_{x_\ell}}A + \frac{1}{\sigma_{x_\ell}^2}B\Big), \quad (25)$$

Substitute our lower bound in Equation 21 on $\sigma_{x_l}^2$ into the above product. Notice that if

$$\sigma_{x_l}^2 \geq \sigma_{x_1}^2 \exp\Big\{S_l\Big\} \quad \text{with} \quad S_l = \sum_{k=1}^{l-1}\Big(\frac{1}{\sigma_{x_k} + 1} + L(\alpha_\ell)\Big) + C, \quad (26)$$

Thus, we have the upper bounds on the inverses:

$$\frac{1}{\sigma_{x_l}} \leq \frac{1}{\sigma_{x_1}}\exp\Big\{-\frac{S_l}{2}\Big\} \quad \text{and} \quad \frac{1}{\sigma_{x_l}^2} \leq \frac{1}{\sigma_{x_1}^2}\exp\Big\{-S_l\Big\}. \quad (27)$$

Thus a valid lower bound for the UP norm is

$$UP\Big(\Big\|\frac{\partial y_L}{\partial x_1}\Big\|_2\Big) \geq \prod_{l=1}^{L-1}\Big(1 + \frac{A}{\sigma_{x_1}}\exp\Big\{-\frac{1}{2}\Big[\sum_{k=1}^{l-1}\Big(\frac{1}{\sigma_{x_k} + 1} + L(\alpha_\ell)\Big) + C\Big]\Big\}$$
$$+ \frac{B}{\sigma_{x_1}^2}\exp\Big\{-\Big[\sum_{k=1}^{l-1}\Big(\frac{1}{\sigma_{x_k} + 1} + L(\alpha_\ell)\Big) + C\Big]\Big\}\Big). \quad (28)$$

Similar to above, assuming that for each layer and based on Equation (24), we have:

$$\sigma_{x_\ell}^2 \le \sigma_{x_1}^2 \exp\left\{\sum_{k=1}^{\ell-1} \frac{1}{\sigma_{x_k}} + M(\alpha_\ell) + C\right\}. \tag{29}$$

So we have:

$$\frac{1}{\sigma_{x_\ell}} \ge \frac{1}{\sigma_{x_1}} \exp\left\{-\frac{1}{2}\left[\sum_{k=1}^{\ell-1} \frac{1}{\sigma_{x_k}} + M(\alpha_\ell) + C\right]\right\}, \tag{30}$$

Hence the UP product is upper bounded by

$$UP\left(\left\|\frac{\partial y_L}{\partial x_1}\right\|_2\right) \le \prod_{\ell=1}^{L-1} \left(1 + \frac{A}{\sigma_{x_1}} \exp\left\{-\frac{1}{2}\left(\sum_{k=1}^{\ell-1} \frac{1}{\sigma_{x_k}} + M(\alpha_\ell) + C\right)\right\}\right.$$
$$\left. + \frac{B}{\sigma_{x_1}^2} \exp\left\{-\left(\sum_{k=1}^{\ell-1} \frac{1}{\sigma_{x_k}} + M(\alpha_\ell) + C\right)\right\}\right]). \tag{31}$$

### D.1 Upper bound of $\mathrm{UP}(\cdot)$

Assume that the $M(\alpha_l)$ is bounded and will not change so much. Inserting this into (31), thus, in the case of constant $\sigma$ and layer-independent $M(\alpha)$, we obtain the explicit bound

$$UP\left(\left\|\frac{\partial y_L}{\partial x_1}\right\|_2\right) \le \exp\left\{\frac{A}{\sigma} \exp\left\{-\frac{1}{2}(M(\alpha) + C)\right\} \frac{1 - \exp\left\{-\frac{L-1}{2\sigma}\right\}}{1 - \exp\left\{-\frac{1}{2\sigma}\right\}}\right.$$
$$\left. + \frac{B}{\sigma^2} \exp\left\{-\left(M(\alpha) + C\right)\right\} \frac{1 - \exp\left\{-\frac{L-1}{\sigma}\right\}}{1 - \exp\left\{-\frac{1}{\sigma}\right\}}\right\}. \tag{32}$$

It is easy to verify:

$$\frac{1 - \exp\left\{-\frac{L-1}{2\sigma}\right\}}{1 - \exp\left\{-\frac{1}{2\sigma}\right\}} = O\left(\min\{L, \sigma\}\right) \tag{33}$$

and similarly

$$\frac{1 - \exp\left\{-\frac{L-1}{\sigma}\right\}}{1 - \exp\left\{-\frac{1}{\sigma}\right\}} = O\left(\min\{L, \sigma\}\right). \tag{34}$$

Plug Equations (33) and (34) into the bound. We deduce

$$UP\left(\left\|\frac{\partial y_L}{\partial x_1}\right\|_2\right)$$
$$\le \exp\left\{\frac{A}{\sigma} \exp\left\{-\frac{1}{2}(M(\alpha) + C)\right\} O\left(\min\{L, \sigma\}\right) + \frac{B}{\sigma^2} \exp\left\{-(M(\alpha) + C)\right\} O\left(\min\{L, \sigma\}\right)\right\}$$
$$= \exp\left\{O\left(\frac{A}{\sigma} \exp\left\{-\frac{1}{2}(M(\alpha) + C)\right\} \min\{L, \sigma\}\right) + O\left(\frac{B}{\sigma^2} \exp\left\{-(M(\alpha) + C)\right\} \min\{L, \sigma\}\right)\right\}.$$
$$= \exp\left\{O\left(\exp\left\{-\frac{1}{2}(M(\alpha))\right\} \frac{\min\{L, \sigma\}}{\sigma}\right)\right\}$$
$$\tag{35}$$

**D.2  Lower bound for** $\mathrm{UP}(\cdot)$

Assume that the $L(\alpha_l)$ is bounded and will not change so much. Inserting this into (28), thus, in the case of constant $\sigma$ and layer-independent $M(\alpha)$, we can obtain the explicit bound.

Assume with $D = \frac{1}{\sigma+1} + L_{(}\alpha)$, For each term in the product (with index $l$), define

$$F(l) = 1 + \frac{A}{\sigma} \exp\left\{-\frac{1}{2}\Big[(l-1)D + C\Big]\right\} + \frac{B}{\sigma^2} \exp\left\{-\Big[(l-1)D + C\Big]\right\}. \tag{36}$$

Taking the logarithm of the whole product, we get

$$\sum_{s=0}^{L-2}\left\{\frac{A}{\sigma}\exp\left\{-\frac{1}{2}(sD + C)\right\} + \frac{B}{\sigma^2}\exp\left\{-(sD + C)\right\} + O\Big(\exp\{-sD - C\}\Big)\right\}. \tag{37}$$

Each of the sums over $s$ is a geometric series. For instance,

$$\sum_{s=0}^{L-2}\exp\left\{-\frac{1}{2}(sD + C)\right\} = e^{-C/2}\sum_{s=0}^{L-2}\left(e^{-D/2}\right)^s = e^{-C/2}\frac{1 - e^{-(L-1)D/2}}{1 - e^{-D/2}}, \tag{38}$$

and similarly for the second term. Notice that when $e^{-D/2} < 1$ the whole sum (even as $L \to \infty$) is bounded by a constant. Exponentiating, we can write:

$$UP\left(\left\|\frac{\partial y_L}{\partial x_1}\right\|_2\right) \geq \exp\left\{\frac{A}{\sigma}e^{-C/2}\frac{1}{1 - e^{-D/2}} + \frac{B}{\sigma^2}e^{-C}\frac{1}{1 - e^{-D}} + O(1)\right\}. \tag{39}$$

similar to the above process, we can get:

$$UP\left(\left\|\frac{\partial y_L}{\partial x_1}\right\|_2\right) \geq \exp\left\{O\left(\frac{1}{\sigma}\frac{1}{1 - e^{(-\frac{1}{\sigma+1} + L(\alpha))/2}} + 1\right)\right\}. \tag{40}$$

$\square$

# E  License

We provide licenses of assets used in our paper, including model, dataset and code.

- C4 dataset: This is the dataset we used in our pretraining experiments. It's released under the Open Data Commons License Attribution family License.
- Commonsense 170K dataset: This is the dataset we used for supervised fine-tuning, released under MIT License.

