# OpenReview forum: "GPAS: Accelerating Convergence of LLM Pretraining via Gradient-Preserving Activation Scaling"
_NeurIPS.cc/2025/Conference — NeurIPS 2025 poster_

### Official Review · Reviewer_aEp8 · 2025-06-13

**Clarity:** 3
**Significance:** 2
**Originality:** 3
**Rating:** 5
**Confidence:** 2

**Summary:**

The paper proposes Gradient-Preserving Activation Scaling (GPAS), an adaptation to Pre-LayerNorm used in LLMs. Pre-LN has an exponential growth in activation variance across layers, which affects the ability of deeper layers to learn effectively. GPAS scales activations, but keeps their gradients unchanged to avoid the vanishing gradient problem.

**Questions:**

### Questions:
* Did you consider the original LLaMA model in your comparisons (e.g., Table 2 and 3)?
* Would you expect higher gains in performance in larger/deeper model ?

### Typos:
* L184: across across
* L247,248: correct to "while,..they fail"

**Ethical Concerns:**

["NO or VERY MINOR ethics concerns only"]

**Final Justification:**

I think the paper addresses an important problem, is well-written and shows good empirical results. I raised minor concerns regarding the generalization to larger model, and these were partially addressed in the rebuttal. I believe it is impossible to evaluate on very large models, as this requires huge computational costs. That said, I must admit that this is not my area of expertise and it is possible that I missed something.

**Limitations:**

yes

**Quality:**

3

**Strengths And Weaknesses:**

### Strengths:
* The paper addresses an important problem with potential high impact and is well-written
* The experiments include several layer normalization schemes, and the evaluation shows consistent gains
* The evaluation considers  pre-training performance (perplexity) and performance on down-stream tasks


### Weaknesses:
* It is not clear if the gains in performance are significant and whether they generalize to larger architectures (also mentioned by the authors in Limitations)

---

> ### Author Rebuttal · Authors · 2025-07-31
>
> Thank you for your constructive feedbacks and thoughtful comments! We have proofread and polished our writing following your suggestions. For your concerns, we will answer them one by one.
>
> >W1. It is not clear if the gains in performance are significant and whether they generalize to larger architectures (also mentioned by the authors in Limitations)
>
> Thank you for mentioning this limitation. We further experimented on 7B parameter models for Pre-LN and Pre+GPAS before the rebuttal period, and GPAS showed even more gains in perplexity compared to smaller scales. We used a learning rate of 3e-4, gradient clipping of 1.0, and trained for 40K steps on 16B tokens, before we stopped the training due to budget constraint. The evaluation perplexities at step 40K are 15.27 and 13.82 for Pre-LN and Pre+GPAS, respectively. The evaluation loss across training steps are shown in the following table:
>
> |Training Step|5K|10K|15K|20K|25K|30K|35K|40K|
> |-|-|-|-|-|-|-|-|-|
> |Pre-LN (7B)|3.854|3.191|2.982|2.886|2.826|2.784|2.752|2.726|
> |Pre+GPAS (7B)|3.844|3.078|2.87|2.777|2.72|2.681|2.651|2.626|
>
> >Q1. Did you consider the original LLaMA model in your comparisons (e.g., Table 2 and 3)?
>
> Thank you for mentioning this. Our main experiments are based on the original LLaMA architecture, specifically its Attn/FFN implementations. We did not compare with the official LLaMA model checkpoints released by Meta [1], since their scale of parameter and data is different from our setting. We have polished our writing of experiment setup in Section 4, so that readers will be more clear about the architectures and checkpoints used.
>
> >Q2. Would you expect higher gains in performance in larger/deeper model ?
>
> Based on our experiments with 7B parameter models, the gains in perplexity are indeed more significant than the 1B setting. When trained for 40K steps, Pre-LN and Pre+GPAS reached perplexities of 15.27 and 13.82, a 1.45 gain in perplexity for GPAS, which is higher than the 0.67 gain in 1B Pre-LN.
>
> [1] Hugo T., Louis M., Kevin S. et al, 2023. Llama 2: Open Foundation and Fine-Tuned Chat Models. arXiv preprint arXiv: 2307.09288

---

> > ### Comment · Reviewer_aEp8 · 2025-08-04
> >
> > I thank the authors for answering my questions. I'll keep my score.

---

### Official Review · Reviewer_XJJw · 2025-06-26

**Clarity:** 3
**Significance:** 2
**Originality:** 3
**Rating:** 4
**Confidence:** 4

**Summary:**

This paper studies the exponential activation variance problem in Pre-LN Transformers. The authors propose GPAS, a simple yet effective method that scales down activations in the forward pass while maintaining gradient magnitudes intact. Experiments conducted across various model sizes (71M to 1B parameters) and architectures (Pre-LN, Sandwich-LN, Post-LN, DeepNorm, Mix-LN, and LNS) demonstrate that GPAS consistently enhances both training stability and model performance.

**Questions:**

- I feel that learning \alpha in GPAS is not straightforward. How do you set your hyperparameters for learning \alpha?
- What is the intuition behind GPAS resulting in larger gradient norms across layers?
- We can control the activation variance of each layer to be at a similar magnitude at initialization. Specifically, this can be achieved by adjusting the initialization standard deviation of weight matrices and using modified residual connections. For instance, some works use: x^{l+1} = 1/L f(x^{l}) + (1- 1/L) x^{l}.
  - Would GPAS still be effective in this setting?
  - If it remains effective, could you explain why?
- I am confused about why "exponential growth in activation variance across layers" leads to deeper layers contributing less to learning. I don't clearly see the direct connection between activation variance and the abstract notion of "contribution," and I think merely observing training/validation loss insufficient.
  - Which experiments in your paper explicitly support this claim, or could you provide further justification?
  - How do you quantify or characterize the contribution of individual layers (especially deeper layers) to the overall learning process of the model?

**Ethical Concerns:**

["NO or VERY MINOR ethics concerns only"]

**Final Justification:**

Based on the authors' reply, I did gain some intuition about language model pretraining. However, my concerns have not been fully addressed. Therefore, I have decided to increase my score from 3 to 4.

**Limitations:**

Yes

**Quality:**

3

**Strengths And Weaknesses:**

Strengths:
- The proposed GPAS method is both simple to implement and broadly applicable to various transformer architectures (pre-LN, sandwich-LN, etc.)
- The performance gain is large. The ablation study is insightful.


Weaknesses:
- Although acknowledged by the authors, the large gradient norm led by GPAS might cause instability and present issues in some training scenarios. In such settings, a gradient clipping algorithm might be needed.
- The model scale up to 1B is good, but the data size is relatively small. I suggest you conduct your experiments with at least a 30B token budget.
- In fact, the gain of GPAS is a bit bigger than I expected. To truly show the effectiveness of the GPAS technique, experiments on more datasets are needed. I strongly recommend the authors to try your GPAS on recently proposed datasets such as DCLM or fineweb, and more tokens.

---

> ### Author Rebuttal · Authors · 2025-07-31
>
> We sincerely appreciate your detailed review and constructive feedbacks. We will address your concerns one by one.
>
> >W1. large gradient norm led by GPAS might cause instability...
>
> Thank you for mentioning this point. We would like to clarify that it is the single gradient spike in warmup stage that led to instability (which only happened in one of the runs), not the large gradient norm itself. Nevetheless, as validated by our 7B experiments, simply applying gradient clipping fixes this potential issue, which we will detail below. Moreover, for the 1B DeepNorm experiment, applying GPAS prevents loss divergence, which shows GPAS introduces stability.
>
> We further tested GPAS on 7B Pre-LN before the rebuttal period. We used a learning rate of 3e-4, gradient clipping of 1.0, and trained for 40K steps on 16B tokens, before we stopped the training due to budget constraint. The evaluation perplexities at step 40K are 15.27 and 13.82 for Pre-LN and Pre+GPAS, which shows more significant gains from GPAS than the 1B setting. The evaluation loss across training steps are shown in the following table:
>
> |Training Step|5K|10K|15K|20K|25K|30K|35K|40K|
> |-|-|-|-|-|-|-|-|-|
> |Pre-LN (7B)|3.854|3.191|2.982|2.886|2.826|2.784|2.752|2.726|
> |Pre+GPAS (7B)|3.844|3.078|2.87|2.777|2.72|2.681|2.651|2.626|
>
> >W2. The model scale up to 1B is good, but the data size is relatively small. I suggest you conduct your experiments with at least a 30B token budget.
>
> We agree that the data scale is relatively small by today's standards. We had to compromise due to limited time and compute budget. However, this limitation does not invalidate our results. Moreover, we further scale up the experiments to 7B parameters and found consistent gains from our proposed method (please see our response to W1). Finally, the loss gap between GPAS and baseline models tend to stabilize after initial stage of training. This means if we extend to train on more tokens, the gains of GPAS will likely remain consistent.
>
> >W3. recommend to try GPAS on recently proposed datasets and more tokens.
>
> We understand that training on more diverse datasets as well as more tokens will further validate our proposed method. Although we cannot rerun our experiments on more datasets due to budget constraint, we believe that this limitation is minor and does not affect our claims.
>
> >Q1. How do you set your hyperparameters for learning \\alpha?
>
> Thank you for raising this question. We set the initial value of $\alpha_l$ to be zero, which corresponds to not scaling activations. There are no other hyperparameters regarding the GPAS operation. As for how GPAS learns $\alpha_l$ during training, we provided a brief explanation in Section 3. We elaborate on this point as follows:
> - Within the same sub-layer, GPAS scales activations after adding Attn/FFN output and the shortcut.
> - When looked across adjacent layers, GPAS can be viewed as scaling the shortcut branch only, since the LayerNorm before Attn/FFN cancels out scaling from the previous layer.
> - By learning to scale the shortcut branch, GPAS is effectively balancing information from Attn/FFN output and the shortcut itself. We believe this is one of the reasons GPAS learns $\alpha_l$ automatically. This intuition also corresponds to the reason why large activation variance is harmful—it makes Attn/FFN outputs overpowered by the shortcut, essentially ignoring their contributions.
>
> >Q2. Intuition behind GPAS resulting in larger gradient norms across layers?
>
> Thank you for pointing this out. We've added a brief explanation in the method section, as well as in our theoretical analysis. From an experimental point of view, GPAS introduces larger gradients for the following two reasons:
> - When we apply GPAS without using stop gradient operator (Equation 10), the layer-wise gradients are of the same magnitude as the baseline model, while the activations are scaled down to a similar degree as standard GPAS with stopgrad. This means that **naively scaling down activations does not change layerwise gradient magnitude**, compared to baseline model with no scaling.
> - Unlike the naive scaling above, GPAS does not scale down backward gradients while scaling down forward activations. This means that during backpropagation, **GPAS gradients are free from the scaling factor aggregated across layers**. Although each layer only introduces a downscaling factor of 0.9~1.0, these factors aggregate across layers, and eliminating them leads to larger gradients of GPAS.
>
> >Q3.
> > - Does GPAS work with scaled initialization?
> > - Does GPAS work with  $x^{l+1} = 1/L f(x^{l}) + (1- 1/L) x^{l}$?
>
> Thank you for raising this question. We will answer your questions one by one.
> - Whether GPAS works with scaled initialization:
>     - We tested Mitchell initialization with Pre-LN (proposed in OLMo [1]), and apply GPAS using Equation 2. **The final evaluation perplexities are listed in the following table**. "Default" refers to Pre-LN with default init, where each weight init uses the same std range:
>
>     |Method|Default|Mitchell|Mitchell+GPAS|
>     |------|------|------|------|
>     |71M|34.23|34.01|32.95 (-1.06)|
>     |350M|21.35|20.49|19.79 (-0.70)|
>
>     - Why GPAS works with Mitchell initialization: After inspecting training statistics, we found that while Mitchell init introduces lower layerwise activation variance (0 ~ 80) compared to default init (0 ~ 120), the layerwise variance is still exponentially growing. Mitchell+GPAS steers layerwise variance to be more evenly distributed (0 ~ 50), as opposed to growing monotonically with layer index.
>     - LayerNorm-Scaling is essentially a scaled initialization of Pre-LN, specifically it initializes LayerNorm weights as $1/\sqrt l$ instead of $1$ ($l$ being the layer index). According to our main experiments (Table 2), LNS+GPAS achieves similar improvement in perplexity compared to Pre-LN.
>     - DeepNorm also uses scaled initialization in Attn/FFN weights, and GPAS provides notable performance gain.
>
> - Whether GPAS works with $x^{l+1} = 1/L f(x^{l}) + (1- 1/L) x^{l}$:
>
>     - We'll name this method "**ScaleRes**" for the sake of discussion, and assume $L$ is the total number of layers. We'll apply GPAS to $x^{l+1}$ the same way as Equation 2, and name it as "**ScaleRes+GPAS**". The table below shows result on 350M models. It seems ScaleRes does not perform as well as Pre-LN. Nevertheless, applying GPAS still offers a 0.84 decrease in evaluation perplexity.
>
>     |Method|Pre-LN|ScaleRes|ScaleRes+GPAS|
>     |---|---|---|---|
>     |Perplexity|21.35|23.22|22.38 (-0.84)|
>
>     - Why GPAS works in this setting: we believe that scaling $f(x_l)$ by $1/L$ is too aggressive, making signals from Attn/FFN even more overpowered by the shortcut branch. In this case, GPAS learns to scale down the activations and effectively scales down the shortcut branch, making Attn/FFN outputs less overpowered, and increases their contribution to learning.
>
> >Q4. Explain the connection between activation variance and the abstract notion of "contribution"
> >- experimental support or further justification?
> >- quantification for layerwise contribution?
>
> We apologize for not being more elaborate when motivating our method. We've polished our writing in Section 1 and 2 to better motivate our work. For the question that you raised, we will answer them one by one.
> - **Connection between "exponential growth" and "deeper layers contributing less"**: In Pre-LN, since the input to Attn/FFN is normalized, the output of Attn/FFN is usually stable in scale (at least during initial training). However, the shortcut branch is not normalized and its variance accumulates and overpowers the signal from Attn/FFN. Thus Attn/FFN would have to learn to scale up their outputs to make meaningful transformation to the shortcut. Ultimately, this causes Attn/FFN's output to also grow with layer depth, despite their input being normalized. This means that **part of the compute is simply dedicated to learning to "scale up" the normalized inputs**, leading to inferior learning efficiency. The learning outcome is also inferior, because Attn/FFN still tend to have much smaller outputs than the shortcut branch, thus **their signal is still overpowered**, limiting their ability to transform hidden states.
> - **Justification for the above claim**: In early stages of our study, we quantified and visualized the above phenomenon by calculating layerwise Jacobian. We found that **deeper layers' Jacobian tend to be close to identity matrix**, which means deeper layers hardly transforms hidden states. But due to the page limit, and that previous works (Mix-LN, LNS, ...) already provide comprehensive analysis of this phenomenon, we did not include our observations in the paper. We will elaborate on this phenomenon in Section 1 to better motivate our work in the updated manuscript.
> - **Quantification for layerwise contribution**: We've added a "Layer Importance Analysis" subsection to Section 5. We quantify layer importance of layer $l$ as the **difference in average benchmark score after removing that layer**, which is an established practice adopted by previous works (Mix-LN, LNS, ...). This is implemented via directly feeding the output of layer $l-1$ to the input of layer $l+1$, essentially ignoring the Attn/FFN outputs of layer $l$. The results show that deeper layers of Pre+GPAS still contribute as much as previous layers, while some layers of vanilla Pre-LN are even harmful to performance. The performance delta of removing certain deep layers (1B model, 24 layers in total) are shown in the following table. A ***positive delta*** means benchmark score increases when removing the corresponding layer, which is undesirable.
>
> |Layer Index|17|18|19|20|21|22|23|24|
> |-|-|-|-|-|-|-|-|-|
> |Pre-LN|***0.42***|***0.19***|***0.02***|***0.44***|-0.69|***0.6***|***0.37***|-0.59|
> |Pre+GPAS|-1.22|-0.86|-1.19|-0.22|-0.64|-0.87|-1.99|-2.31|
>
> [1] Team OLMo et al, 2025. 2 OLMo 2 Furious. arXiv preprint arXiv:2501.00656

---

> ### Comment · Reviewer_XJJw · 2025-08-04
>
> Thank you for your response.
>
> The rebuttal addressed some of my concerns, and I learned a lot from your explanations. I still have some further questions:
>
> > About how GPAS works: activation norm
>
> Firstly, I believe the authors attribute GPAS's effectiveness to decreasing the layerwise activation norm, specifically in the Pre-LN architecture. Your explanation also makes sense to me.
>
> However, certain techniques can also mitigate this problem at initialization. For example, fan-in initialization combined with a scaled residual can **collectively** maintain a **constant** magnitude of activation norm across layers at initialization (instead of 0 - 80). In your Figure 3, we clearly see that the layerwise activation norm is unbalanced -- or more precisely, increasing -- at initialization.
>
> In my review, I suggested trying these initialization techniques, and in your rebuttal, you demonstrated GPAS's effectiveness. However, I would like to know if the variance of activation norms across layers still becomes large as training proceeds. If so, does GPAS still operate through the same mechanism?
>
> I agree that, under default initialization (constant std, such as 0.02), the shortcut signal can overpower deeper layers since the activation norm increases with layer depth, leading to more significant issues in deeper layers. However, with scaled residual connections using convex addition, the activation norm "should" remain stable across layers. At this point, we don't need a direct comparison between scaled residual and standard Pre-LN.
>
> Overall, my question is: if we explicitly control the activation norm so that it does not explode with depth, will GPAS still operate through the same mechanism?
>
> > About how GPAS works: hyperparameter tuning
>
> I am also concerned about hyperparameters. Specifically, is it possible that GPAS's improvements might diminish if both the baseline and GPAS hyperparameters are well tuned? The optimal hyperparameters might change when applying GPAS, and the settings from previous works might not be directly relevant.
>
> Reviewer odJN shares this concern.
>
> Additionally, I strongly agree with Reviewer MBu7's comment:
> "Pretraining models is expensive and there are a lot of papers claiming their method significantly improves the training. If I'm honest, they generally do not in practice, and people working on production models often tend to end up."
>
> ---
>
> Based on the authors' reply, I did gain some intuition about language model pretraining. However, my concerns have not been fully addressed. Therefore, I have decided to increase my score from 3 to 4.

---

> > ### Author Response · Authors · 2025-08-04
> >
> > Thank you for the followup comments and for raising the score! For your additional questions, we provide the following clarifications:
> >
> > > However, I would like to know if the variance of activation norms across layers still becomes large as training proceeds. If so, does GPAS still operate through the same mechanism?
> >
> > For Pre-LN (both default init and scaled init), the layerwise activation variances usually start at a small scale (the first \~1K training steps). As training progresses, variances slowly increase to a stable scale (e.g. around 40K steps in Fig 3a), then remain at that scale for the rest of training. Since the variances show similar dynamics, we believe GPAS operates with the same mechanism.
> >
> > > if we explicitly control the activation norm so that it does not explode with depth, will GPAS still operate through the same mechanism?
> >
> > Yes, GPAS still operates with the same mechanism.
> >
> > First, we would like to clarify that stable activation variance is only one part of the picture. Another important aspect is the **mixing ratio between Attn/FFN outputs and the shortcut**, which affects the transformation on hidden states.
> > - For instance, LNS [3] already maintains stable activation variances through depth-dependent scaling. GPAS enhances LNS with an adaptive scaling scheme that dynamically adjusts the mixing ratio between Attn/FFN outputs and the shortcut without scaling gradients.
> > - In vanilla Pre-LN where variances explode with layer depth, GPAS scales down activation which essentially scales down the shortcut, so that the mixing ratio is improved.
> >
> > Therefore, in both cases, GPAS operates through the same mechanism—it adaptively balances information from Attn/FFN and the shortcut, so that the transformation on hidden states is more effective.
> >
> > > About hyperparameter tuning.
> >
> > Thank you for raising this question. We directly adopted hyperparameters from previous works [1,2,3] which are already tuned for the baseline models, and did not tune specifically for GPAS. We believe directly adopting their settings does not benefit our approach. We will further investigate the effects of hyperparameters as part of our ongoing research efforts.
> >
> > Once again, we thank you for your time and constructive feedbacks. We do find the points you raised very interesting and will conduct further research on those topics.
> >
> > [1] Zhao, J., Zhang, Z., Chen, B., Wang, Z., Anandkumar, A. and Tian, Y., 2024. Galore: Memory-efficient llm training by gradient low-rank projection. ICML 2024.
> >
> > [2] Li, P., Yin, L. and Liu, S., 2024. Mix-ln: Unleashing the power of deeper layers by combining pre-ln and post-ln. ICLR 2025.
> >
> > [3] Sun, W., Song, X., Li, P., Yin, L., Zheng, Y. and Liu, S., 2025. The curse of depth in large language models. arXiv preprint arXiv:2502.05795.

---

### Official Review · Reviewer_MBu7 · 2025-07-02

**Clarity:** 2
**Significance:** 3
**Originality:** 2
**Rating:** 5
**Confidence:** 3

**Summary:**

This work builds on the observations that increasing the depth of models has come with diminishing returns by proposing an amendment to existing normalisation schemes. This modification, GPAS, aims at rescaling the variance of activations without affecting the scale of the gradients. The authors evaluate adding their technique to several normalisation schemes empirically, and explore the effect of GPAS on the variance of activations and the norm of the gradients, along with other ablations on the choices made in the design of GPAS. The authors also propose a theoretical analysis of how GPAS affects the norm of the gradients.

**Questions:**

**Questions**
1. “Moreover, by scaling $x’_{l+1}$ , $\alpha_l$ controls the mixing ratio between residual [...] effectively balancing information from these 2 variables” I am not sure I fully understand this sentence. How does $\alpha_l$ intervene in the mixing ratio of the residual and the output of Attn or the FFN? My understanding of eq. 2 is that $\alpha_l$ affects both the residual and the output of attention/FFN equally since it’s applied after they are added together (in the form of a scaling factor $1-\text{SiLU}(\alpha_l)$).
2. p3: did you experiment with applying GPAS to the entire argument of LN in eq 7 and/or replacing $x_l$ with $x’_l$ in the argument of $f$ in the same equation? Since you mention experimenting to find how to best apply GPAS in the post-LN / DeepNorm cases, negative results would also be interesting to readers of the paper experimenting with architectural changes.
3. “We believe that the combination of larger weights and stable activation variance across layers enables the network to tolerate larger gradients without crashing, which in turn speeds up convergence. “ This conclusion about GPAS not crashing would need to be explained. If anything, the previous section showed that GPAS made training less stable.
4. Lemma 1: there are multiple weights involved at layer l: $W_Q, W_K, W_V, W_O$ and fully connected layers weights. There is also the LN weight. So what is $W_l$? The concatenation of all these weights?

**Suggestions**
1. p4: “We did not report Post + GPAS [...]” I suggest adding that information to the Tab 2 caption too so readers may get that information at a glance
2. Tab 3: In my experience, the MMLU and WinoG results are all too close to random chance to not be within variance of said random chance. For example, when pretraining, MMLU tends to fluctuate within a few percent of 25% until it “takes off” and starts making consistent progress. As far as I am concerned this doesn’t weaken the claim about GPAS based on other evals, so it’s not an issue per se, but I want to draw attention to it in case you may want to allude to that in your discussion to anticipate readers pointing that out.
3. Tab 4: “scaled SiLU” is commonly referred to as “SwiGLU”, I recommend using that term.
4. p3: did you use any kind of weight decay? If so, please add it in the paper.
5. Theorem 1: I suggest avoiding the simultaneous use of $L$ as the layer count and as a lower bound function for clarity
6. More of a suggestion, but I heavily recommend configuring your bibliography-style to one supporting hyperlinks for your references. For example, “arXiv preprint arXiv:2407.21783” in the first reference being clickable would enhance readability.

**Typos**
1. “in LLM pertaining” (end of sec 4.1) should be “pretraining”
2. Tab 3: For WinoG, the 2nd best performance is Pre+GPAS (52.25), not LNS (52.09)
3. Lemma 1 would benefit from rephrasing. For example, the first sentence needs rephrasing and is potentially missing a reference to equation 2, or “Let the upper bound for this 217 norm denoted by UP($\cdot$)” needs to be “be denoted”

**Ethical Concerns:**

["NO or VERY MINOR ethics concerns only"]

**Final Justification:**

**Post-rebuttal**
The authors will provide clarifications and fixes accordingly. I count on them to rerun experiments with gradient clipping for the camera ready version, which should not change the results significantly but is conceptually important to reflect realistic setup and address instabilities observed. I also count on them to remove claims that their method improves stability. I think it's great that they indicated the negative result for readers. I increased my score from 3 to 5.

**Limitations:**

Yes.

**Paper Formatting Concerns:**

No.

**Quality:**

2

**Strengths And Weaknesses:**

**Summary of strengths**
1. The proposed modification is simple and is computationally cheap, and aims to solve the important issue of suboptimal learning in deep layers of LLMs
2. The authors discuss design ablations and negative results, which will help the community build upon this work
3. The authors’ experiments show that at a small scale (of models and training flops), GPAS helps and the claim that GPAS helps normalising the variance’s behaviour across layers is supported experimentally. The experimental work is easy to follow.

**Summary of weaknesses**

Generally, the paper has 3 kinds of weaknesses that would be essential to fix before I would consider it ready for publication:
1. Claims: Issues with clarity of claims about how GPAS works or performs (notably, being invalidated by experiments, e.g. re: stability or not affecting gradients).
2. Instability & gradient clipping: I am glad that the authors raise the point of stability and added Fig 4. A key aspect of training foundation models is balancing the speed of learning with stability, with the latter being increasingly harder to achieve as the model or token budget is scaled up. However, GPAS introduces instability, even at a very small scale (<= 1B params with <= 10B tokens). As the authors point out, they did not use gradient clipping, except for a model that could not recover from instability at all otherwise, even though it is a default when training LLMs. In my opinion, and you may anticipate this sentiment to be shared by many readers, it is vital that the authors reevaluate their method across the paper with gradient clipping, especially because the experiments do not include models larger than 1B parameters – which are typically stable to train – and already see stability issues. Since you mention requiring 2-3 days per 1B model, I think you could focus on Pre and Pre+GPAS so you can update Fig 4, and retrain + reevaluate the models after the rebuttal period. Again, as the authors claim in the motivation section, experimenting with foundation models is very costly, so I would be reluctant to use compute on a method introducing instabilities – in other words, you cannot leave a fix as a hypothetical.
3. Theoretical analysis: the theoretical analysis needs significant attention due to phrasing issues and/or incorrect claims. Moreover, it needs to be more stand-alone, since at times, it seems necessary to read reference [4] (the curse of depth paper) to understand some arguments (which are stated in [4] but not so clearly in the present work).

**More feedback**
1. Minor, but missing citation of *The Unreasonable Ineffectiveness of the Deeper Layers* by Gromov et al. from early 2024 when motivating your work, which was one of the first works noticing the depth issue.
2. p2: Giving a little bit more info about Mix-LN is important; specifically explaining how it is not just using both Pre-LN and Post-LN everywhere in the architecture (which would predate Mix-LN, Gemma 2 for example introduced that before).
3. p3: $\beta$ is introduced in the text but does not appear anywhere in the DeepNorm equations. A potential solution is to subscript $f$ by $\beta$ and point to something more specific in the appendix. This is important because it’s not clear what happens to $\beta$ with GPAS. Since the text only mentions adjusting $\alpha$, I assume $\beta$ is left untouched (i.e. keeps being used).
4. p3: Eq 6 and 7 show how GPAS is used with DeepNorm. While I assume it corresponds to $\alpha=1$ in eq. 7, I believe it would be important to either explicitly mention it near eq. 7, or to have an equation for PostLN+GPAS to avoid any confusion (and clarify in the text whether the sentence “Empirically, we found applying GPAS after LayerNorm does not bring performance gain” applies solely to DeepNorm or also to Post-LN).
5. p4: “Details on how 121 we augment each baseline are discussed in Section 3” Section 3 does not mention LNS and LNS+GPAS. It is worth expanding on it in section 3 by giving the equations as done for the other baselines (especially since LNS+GPAS is arguably the best performing baseline in Tab 2).
6. Lemma 1 / theorem 1: how can $x_l$ and $x’_l$ be independent when $x_l$ is a function of $x’_l$ such that $x’_l$ determines the value of $x_l$? You might suppose it for a proof that would make use of eq 2, but this supposition already rules out applying the theoretical results to GPAS since these variables are quite dependent. Reading from the curse of depth paper, their $x_l$ and $x’_l$ clearly are not “mutually independent” and they do not assume non-zero covariance, so I assume the phrasing of your assumption needs to be clarified. That being said, their work uses quite an unrealistic assumption that the output of the attention softmax is equal to $1/n I$. It could be justified by the exploding variance, but I don’t see them mentioning it (and it would change the analysis slightly since that would not be true for a finite and potentially large number of layers).
7. Eq 8: $M$ is never introduced.
8. “This leads to a strictly non-constant lower bound, which in turn ensures that the gradient magnitude retains sufficient variability across layers. In particular, this precludes the possibility of vanishing gradients and encourages meaningful gradient flow during backpropagation.” I may be missing something so correct me if I’m wrong, but Theorem 1’s lower bound is a lower bound on the upper bound of the gradient norm. It is not a lower bound on the gradient norm itself, so it might very well be 0 without violating theorem 1.
9. “ Consequently, the proposed modification mitigates gradient explosion and reduces the occurrence of large loss spikes, thereby enhancing training stability” again, I do not understand the claim as this was not the case empirically, nor does the theory prove this in real world settings due to the assumptions being made as discussed in previous points.
10. The claim about GPAS being an intervention that leaves the gradients intact, that appears to be the main claim as it is underlined on the first page and is mentioned in the abstract, is not correct as shown empirically (and as the authors want to show in the analysis since based on [4], it’s a way to show the intervention on the variance corrects undesirable behaviour observed with the norm of the gradient with respect to the first layer’s parameters). You may want to argue that unlike direct interventions on the gradients, you target a cause of problematic gradients and not just the symptoms.
11. Theorem 1: should the $\alpha$ argument of $M$ and $L$ be indexed by $l$ since $M, L$ are bounds on $1-\text{SiLU}(\alpha_l)$?
12. Theorem 1, as opposed to the curse of depth paper, suggests that the upper bound will not scale as the number of layers increases. But the curse of depth papers argues that this is undesirable as it means that additional layers will act as the identity mapping, or in other words, that deep layers become ineffective. I tend to agree with that desiderata – by the chain rule, a model with very large $L$ should have (sufficiently) large gradients for every layer, and hence the norm of the gradient with respect to the first layer’s parameters should be increasingly large with increasing L, unless a method targeting the gradients is used (here, the authors claim GPAS “leav[es] the backward gradients intact”).

---

> ### Author Rebuttal · Authors · 2025-07-31
>
> We sincerely appreciate your very detailed review and constructive feedbacks! Due to word limit, we kindly acknowledge your following suggestions together, and conduct significant proofreading and polishing accordingly:
> - W3 (Weakness 3)
> - MF1 (More feedback 1), MF2, MF4
> - S1 (Suggestion 1), S3, S5, S6
> - Typos
>
> We will now answer your remaining concerns.
>
> >W1. Further clarify how GPAS works or performs.
>
> - **How GPAS works**. GPAS addresses activation explosion in Pre-LN while preserving gradients. In LNS and DeepNorm, it enhances static scaling with a learnable one that adapts to training dynamics. For Sandwich-LN, it dynamically scales the shortcut branch for better blending. In Mix-LN, it enhances both Pre- and Post-LN layers.
> - **Stability**. Stability is only mildly affected in the 1B Pre-LN run due to a single warmup spike, which gradient clipping resolves (see W2). On 1B DeepNorm, GPAS prevents loss divergence, and on 7B Pre-LN (see W2), it improves convergence without instability. The 7B results will be added in Appendix A.
> - **Gradients**. The claim that GPAS "does not affect gradients" refers to Eq.2—its Jacobian is identity, so it does not scale backpropagated gradients, though it scales down the forward activations. We will clarify this in Section 3. The resulting increase in layerwise gradients arises because:
>     - The "naive" version (Eq.10, without stopgrad) empirically does not change layerwise gradient norm compared to the baseline, but reduces activations similar to Eq.2.
>     - Standard GPAS (Eq.2), avoids gradient downscaling of Eq.10—thus amplifying layerwise gradients relative to naive scaling.
>
> >W2. Instability & gradient clipping.
>
> We agree gradient clipping is generally recommended for LLM training. However, we initially excluded it for the following reasons:
> - Follow settings of prior works (Mix-LN, LNS).
> - Stress-test GPAS under harsher conditions.
> - Gradient clipping has limited impact on small models and does not affect GPAS gains, PPL shown below:
>
> |Method|71M Pre-LN|71M Pre+GPAS|350M Pre-LN|350M Pre+GPAS|
> |-|-|-|-|-|
> |no gradclip|34.23|33.49|21.35|20.35|
> |gradclip (1.0)|33.99|33.33|20.92|19.96|
>
> We also ran 7B pretrain with gradient clipping, confirming that GPAS scales well and remains stable. Although we couldn’t rerun 1B due to crowded server, we tested GPAS on 7B Pre-LN before the rebuttal using LR 3e-4, clipping 1.0, and 40K steps on 16B tokens. At step 40K, Pre-LN and Pre+GPAS achieve perplexities of 15.27 and 13.82, respectively. The full eval loss is shown below:
>
> |Training Step|5K|10K|15K|20K|25K|30K|35K|40K|
> |-|-|-|-|-|-|-|-|-|
> |Pre-LN (7B)|3.854|3.191|2.982|2.886|2.826|2.784|2.752|2.726|
> |Pre+GPAS (7B)|3.844|3.078|2.87|2.777|2.72|2.681|2.651|2.626|
>
> >MF3. $\beta$ should appear in the DeepNorm equations as subscript.
>
> Yes, $\beta$ is left untouched for DeepNorm+GPAS. We've added $\beta$ as a subscript in our updated manuscript.
>
> >MF5. Section 3 does not mention LNS and LNS+GPAS.
>
> Thanks for pointing out. We'll give separate equations in Section 3. The LNS forward pass is $x_{l+1}'= x_l+f(\mathrm{LN}(x_l)/\sqrt{l})$, and LNS+GPAS adds Equation 2 same as Pre-LN.
>
> >MF6. Lem/Thm.1: how can $x_l$ and $x’_l$ be independent...
>
> Thank you for pointing this out. We did not mean $x_l$ and $x'_l$ are mutually independent—only that both are independent of the weight matrix $W_l$. We will clarify this.
>
> You are right that the Softmax output is not uniform—our statement was a simplification. However, in our analysis we focus on the asymptotic analysis of variance of the Softmax. This involves a summation over N terms (N denotes the number of key-value pairs), which are assumed to be independent. When N is large, by the theory of concentration of measure in high dimensions, the variance converges to that of the uniform case. Thus, the non-uniformity does not affect the result. This is also supported by Lem A.1 in the updated Curse of Depth paper.
>
> >MF7. Eq 8: $M$ is never introduced.
>
> You are right that $M$ was never defined. In our notation, $M$ denotes an arbitrary non-zero constant. Equivalently, this means that the stated upper bound actually admits a strictly positive constant lower bound $M>0$. We will correct this in the later version.
>
> >MF8. Thm.1’s lower bound...might very well be 0 without violating Thm.1.
>
> We believe there's a misunderstanding, and we offer the following clarification. Our result establishes a nontrivial range for the upper bound on the gradient norm; it does not assert a direct lower bound on the gradient norm itself. Indeed, the actual gradient norm can become zero after $L$ layers due to cancellations or “ups and downs” in the network. So the lower bound must be zero. However, characterizing how the upper bound behaves is still important: it shows that no matter how deep the network grows, the worst-case gradient magnitude remains confined within a fixed range and does not blow up or vanish exponentially. Moreover, no matter how large $L$ becomes, the upper bound remains a constant—giving rise to the “curse of depth” phenomenon in which deeper layers become entirely ineffective. In our paper, Thm1 can avoid this because the upper bound must be larger than a constant $M$. Hence, characterizing the behavior of this upper bound remains meaningful. Finally, note that loss spikes are directly linked to this gradient upper bound (see the discussion in the Curse of Depth paper).
>
> >MF9. "Consequently...thereby enhancing training stability" again, I do not understand the claim...
>
> We want to clarify that our theoretical analysis is more of a statement about the potential of GPAS to enhance stability, and did not take training dynamics into account (mainly $\alpha_l$ being learnable). A full treatment of training dynamics would indeed require a far more detailed analysis, which is part of our ongoing research efforts.
>
> >MF10. Claim about GPAS leaving gradients intact seems incorrect empirically.
>
> As we have clarified in W1, "leaving the gradient intact" is with respect to the naive scaling case.
>
> >MF11. Thm.1: should the $\alpha$ argument of $M$ and $L$ be indexed by $l$?
>
> Thanks for the question. In our notation, $M(\alpha)$ and $L(\alpha)$ are uniform bounds that apply to all $\alpha_l$ for $l \leq L$. Since these bounds hold simultaneously across all layers, they are not indexed by $l$. We will clarify this point in the revised manuscript.
>
> >MF12. Thm.1's upper bound seems to conflict with the curse of depth paper.
>
> Your point is right, but our results do not conflict with theirs. The Curse of Depth paper shows that the gradient-norm upper bound can remain a fixed constant, which is undesirable. In Lemma 1, We do not say our gradient norm is a constant, but the upper bound of gradient norm must be larger than a constant. In contrast, we tighten the range of that upper bound: instead of a constant lower bound, we show that the lower bound itself grows with depth. At the same time, the upper bound’s growth rate is reduced, so it does not explode too rapidly. As a result, the overall band of possible gradient norms is narrower, which means that, with GPAS, we achieve better performance by avoiding both unbounded growth and constancy of the upper bound.
>
> >Q1. How does $\alpha_l$ affect mixing ratio?
>
> Thanks for the question. While GPAS is applied after adding the outputs of Attn/FFN and the shortcut within the same sub-layer, across adjacent layers it effectively scales only the shortcut branch—since the LayerNorm before Attn/FFN removes any scaling from the previous layer. In this sense, $\alpha_l$ modulates the mixing ratio between the shortcut and the Attn/FFN outputs. We will clarify this interpretation in the revised manuscript.
>
> >Q2. Clarify DeepNorm+GPAS negative trials.
>
> We experimented with applying GPAS to the entire argument of Equation 5, specifically to the $x_{l+1}$ in Equation 5. We did not experiment with combining both approaches. The negative trials are not included due to page limit. We are actively reproducing negative results to be put in the appendix of the revised manuscript.
>
> >Q3. Explain conclusion about GPAS not crashing.
>
> Thank you for raising this point. First we clarify that GPAS does not make training less stable—standard gradient clipping fully resolves the observed issues (see W1 & W2).
> By "tolerate larger gradients" we mean GPAS layers operate with larger gradients than baselines, especially in early layers. "Larger weights" are less sensitive to gradient noise, making them more robust. "Stable activations" refer to the consistent activation magnitudes across layers, as shown in Fig 3b.
>
> >Q4. Lem 1: what's $W_l$?
>
> To clarify, by $W_l$ we refer to a standard weight matrix (exluding LN weights). In our setting, all weight matrices are treated uniformly as i.i.d. with zero mean and variance
> . Therefore, $W_l$ in this context refers to any such weight matrix e.g. $W_Q, W_K, W_V, W_O$. We will clarify this point in the revised manuscript.
>
> >S2. Clarify MMLU and WinoG results.
>
> Thank you for pointing this out. The original MMLU score was due to using an outdated version of `lm-eval`. After updating, the corrected results are:
>
> |Method|MMLU|
> |-|-|
> |Post-LN|22.95|
> |DeepNorm|22.95|
> |DeepNorm+GPAS|26.46|
> |Pre-LN|25.95|
> |Pre+GPAS|26.68|
> |Sandwich-LN|27.29|
> |Sandwich+GPAS|27.41|
> |Mix-LN|26.24|
> |Mix+GPAS|26.23|
> |LNS|26.62|
> |LNS+GPAS|27.81|
>
> This is more aligned with the reported results of Mix-LN [1] and LNS [2].
> We believe the low MMLU and WinoG scores were due to:
> - MMLU being out-of-distribution for our C4 pretraining corpus, which favors everyday knowledge over academic reasoning.
> - Lack of instruction finetuning, which limits performance on instruction-based benchmarks.
>
> That said, our core claims remain intact—on commonsense reasoning tasks closer to C4, GPAS still yields consistent gains.
>
> >S4. Weight decay usage.
>
> Following Mix-LN and LNS, we did not use weight decay for the main experiments.
>
> [1] Mix-LN. 2025.
> [2] LNS. 2025.

---

> ### Comment · Reviewer_MBu7 · 2025-08-03
>
> I thank the authors for their careful rebuttal. I look forward to them adding all clarifications/fixes to the paper.
>
> > its Jacobian is identity, so it does not scale backpropagated gradients, though it scales down the forward activations. We will clarify this in Section 3. The resulting increase in layerwise gradients arises because:
>
> I understand the point that stopgradient means that the GPAS modification e.g. in eq2 does not modify the calculation of the derivative. But gradients are then evaluated as functions of the activations, so since GPAS affects the activations, it does affect the gradients, no? E.g. if we denote GPAS as some function $f(x)$, and the model computes $g(f(x))$, then when calculating the gradient, even if $f'$ is trivial, $g'(f(x))$ will appear and be affected by the activation $f(x)$.
>
> > Follow settings of prior works (Mix-LN, LNS).
>
> I just want to raise a concern that prior work may very well be flawed too. Even peer-reviewed papers. I'd have the same feedback for those papers. My thought process is the following: readers who will benefit the most from such work are likely people pretraining models. Pretraining models is expensive and there are a lot of papers claiming their method significantly improves the training. If I'm honest, they generally do not in practice, and people working on production models often tend to end up... disconnecting from the literature. That's why running the experiments with realistic conditions / setups is so important: it makes it significantly more believable that spending effort and compute on a given method may be worth it because it is closer to their setup.
>
> > Gradient clipping has limited impact on small models and does not affect GPAS gains, PPL shown below:
>
> Performance is not really the main reason (but it is one) for using gradient clipping with AdamW for pretraining; stability is.
>
> > You are right that the Softmax output is not uniform—our statement was a simplification. However, in our analysis we focus on the asymptotic analysis of variance of the Softmax.
>
> I suggest clarifying all such assumptions in your paper, since currently one has to read the appendix of another paper to guess that.
>
> > Stability. Stability is only mildly affected in the 1B Pre-LN run due to a single warmup spike, which gradient clipping resolves (see W2). On 1B DeepNorm, GPAS prevents loss divergence, and on 7B Pre-LN (see W2), it improves convergence without instability. The 7B results will be added in Appendix A.
>
> Coming back to this: the 7B results will be a valuable addition. That being said, in hundreds if not more pretrainings of 1B models, even back then with fp16, I have never seen unstable runs at 1B. So, this is a very big deal/concern to me, and as much as I hate resorting to that argument again, "trust me when I say people pretraining models will think the same". Hence why it's so important for you to rerun experiments with gradient clipping for all. I trust that you will do it for the camera ready version. I also want you to commit to removing claims that their method brings stability to training, because if anything, there is evidence of the contrary. This is important. In anticipation of that, I increase my score from 3 to 5. Again, kudos for highlighting the instable run in the paper, this is really the kind of information that is absurdly important when reading such work but that often gets cut to optimise peer review chances.

---

> > ### Author Response · Authors · 2025-08-03
> >
> > Thank you for your prompt feedback and for raising the score. Your suggestions are greatly appreciated and help us further improve the paper.
> >
> > > But gradients are then evaluated as functions of the activations, so since GPAS affects the activations, it does affect the gradients, no?
> >
> > We fully agree with this argument. While GPAS does not modify the gradient path due to stopgradient, it affects the forward activations, which in turn influence the gradients. We will make this clearer in the revised version and add a brief analysis of this effect.
> >
> > > I suggest clarifying all such assumptions in your paper, since currently one has to read the appendix of another paper to guess that.
> >
> > Thank you for pointing this out. We will carefully review the paper and make all assumptions explicit in the main text, without requiring readers to refer to external sources.
> >
> > > Rerun experiments with gradient clipping. Remove claims about bringing stability to training.
> >
> > We are actively rerunning all experiments with gradient clipping and will include the updated results in the camera-ready version. We also acknowledge signs of instability in certain run and will revise the paper to ensure our claims regarding stability are accurate and consistent with the observed behavior.
> >
> > Once again, we sincerely appreciate your thoughtful and constructive comments. We are committed to addressing these concerns and improving the final version to better serve the community.

---

> > > ### Comment · Reviewer_MBu7 · 2025-08-03
> > >
> > > Thank you for your hard work!

---

### Official Review · Reviewer_odJN · 2025-07-03

**Clarity:** 3
**Significance:** 3
**Originality:** 3
**Rating:** 5
**Confidence:** 3

**Summary:**

The authors propose a simple modification of residual connections in transformers that aims to adjust the scale of intermediate activations without impacting the scale of gradients. They test the new GPAS method by pretraining and finetuning language models. The results show that the method improves performance and decreases the variance of intermediate activations.

**Questions:**

- Is there any negative impact of training speed? I would expect not, but it is always better to ask. I think it would be better to clearly mention this in the paper.
- Since you decided to use a relatively established Commonsense170K dataset/benchmark for evaluation, why not reporting the performance on all datasets in the benchmark?
- Is it correct to the SFT evaluation as zero-shot when the models are directly trained on training samples from the evaluated datasets?
- In the experiments with predefined gates, what is the impact of this setting on the activation and gradient variance?

**Ethical Concerns:**

["NO or VERY MINOR ethics concerns only"]

**Final Justification:**

I'm quite confident with keeping the score given to this paper the same, I believe that 5 (accept) is a fair assessment for the contributions of this paper.

**Limitations:**

yes

**Paper Formatting Concerns:**

No concerns

**Quality:**

3

**Strengths And Weaknesses:**

**Strengths**
- First of all, I really like the simplicity of this method. Researchers often try to impress each other by proposing complicated contraptions, but this is not the case; it tries to solve a complicated problem with a simple fix.
- The authors have tested their method not only on the most popular pre-layernorm architecture but also on four other architectures. The improvement due to their modification seems to be consistent across them.
- The paper is well-written and was easy to understand and follow,
- The empirical analysis that demonstrates the effects of GPAS is extensive and clear.
- The paper also contains detailed ablations studies that experimentally confirm the various choices of the GPAS modification.

**Weaknesses**
- The evaluation of the downstream performance could be improved. For example, the performance of all models on MMLU and WinoGrande is equivalent random guessing (all MMLU results are actually worse than the 25% random baseline). Thus, the advertised improvement on these benchmarks is most likely just random noise. Overall, the improvements are consistent, but small, so adding some confidence information would help to make the evaluation more reliable.
- The mathematical justification in 5.5 does not seem to add much value. If I understand it correctly, it merely shows that the new parameterization has the potential of stabilizing the gradient norms, not that it actually achieves that when the parameters are learned. That would need a much more complicated analysis of the training dynamics.
- The language models used for the empirical evaluation are all under-trained, at least according to the "Chinchilla scaling laws" [1]. I definitely do not think that they are under-trained to a degree that would invalidate the results, but it is still a concern. Modern language models are trained way past the point of "optimal data size", so it is important to test how the proposed method behaves in this regime. Note that that does not necessarily have to be done using a large and expensive language model.
- Similarly, other slightly concerning aspect of the experimental setup is that the hyperparameters were not optimized specifically for each evaluated method. For example, could it be the case that GPAS works better just because of the chosen learning rate but models without GPAS would be equally good when trained with a slightly lower learning rate? I understand that doing a hyperparameter search for every experiment is prohibitively expensive, but it could again be done separately in a small-scale setting.

---

[1] [Training Compute-Optimal Large Language Models](https://arxiv.org/abs/2203.15556)

---

> ### Author Rebuttal · Authors · 2025-07-31
>
> We sincerely appreciate your constructive feedbacks and insightful comments! As for your concerns, we will address them one by one as follows.
>
> >W1. The evaluation of the downstream performance could be improved. For example, the performance of all models on MMLU and WinoGrande is equivalent random guessing (all MMLU results are actually worse than the 25% random baseline). Thus, the advertised improvement on these benchmarks is most likely just random noise.
>
> Thank you for noticing. We believe the MMLU score was caused by using an older version of `lm-eval`. After updating to the latest version, here are the re-evaluated results:
>
> |Method|MMLU|
> |-|-|
> |Post-LN|22.95|
> |DeepNorm|22.95|
> |DeepNorm+GPAS|26.46|
> |Pre-LN|25.95|
> |Pre+GPAS|26.68|
> |Sandwich-LN|27.29|
> |Sandwich+GPAS|27.41|
> |Mix-LN|26.24|
> |Mix+GPAS|26.23|
> |LNS|26.62|
> |LNS+GPAS|27.81|
>
> This is more aligned with the reported results of Mix-LN [2] and LNS [3].
> We summarize the reasons for near random score on MMLU and WinoG as follows:
> - MMLU is out-of-distribution relative to our pretraining corpus. Our models are trained on the C4 dataset, which is generic web corpus. This dataset is good for language modeling and learning everyday knowledge, but not ideal for knowledge/reasoning intensive tasks.
> - Our models have not undergone instruction finetuning (except for Commonsense170K SFT), which further limits their performance on tasks requiring instruction following or academic reasoning.
>
> Nonetheless, this does not undermine our core claims. On Commonsense Reasoning benchmarks—which are more closely aligned with the C4 domain—GPAS still showed consistent and meaningful improvements.
>
> >Overall, the improvements are consistent, but small, so adding some confidence information would help to make the evaluation more reliable.
>
> We fully agree with this point. We will add confidence information in our updated manuscript.
>
> >W2. The mathematical justification in 5.5 does not seem to add much value. If I understand it correctly, it merely shows that the new parameterization has the potential of stabilizing the gradient norms, not that it actually achieves that when the parameters are learned. That would need a much more complicated analysis of the training dynamics.
>
> We admit that the mathematical justification in Section 5.5 is preliminary. Our goal was to offer theoretical insights into how GPAS modulates gradient and activation variance. By leveraging the results from Curse of Depth [3], Theorem 1 and Lemma 1 in our manuscript already provide evidence that GPAS improves gradient propagation—even without an explicit training‐dynamics derivation. A full treatment of the training dynamics would indeed require a far more detailed analysis, which is part of our ongoing research efforts and will be systematically explored in future publications.
>
> >W3. The language models used for the empirical evaluation are all under-trained, at least according to the "Chinchilla scaling laws" \[1\]. I definitely do not think that they are under-trained to a degree that would invalidate the results, but it is still a concern. Modern language models are trained way past the point of "optimal data size", so it is important to test how the proposed method behaves in this regime. Note that that does not necessarily have to be done using a large and expensive language model.
>
> First we want to clarify that our pre-training setting closely follows the setting that are widely used in previous works [1,2,3]. We also agree that the models are under-trained by today's standards. An observation that alleviates this concern, is that the training loss gap between GPAS models and baseline models tend to stabilize after the initial 30% steps, and remain consistent gap for the rest of training. Based on this observation, it is likely that, when continued to trained on more data, the improvements of GPAS over baselines will be similar to the under-trained scenario.
>
> >W4. Similarly, other slightly concerning aspect of the experimental setup is that the hyperparameters were not optimized specifically for each evaluated method. For example, could it be the case that GPAS works better just because of the chosen learning rate but models without GPAS would be equally good when trained with a slightly lower learning rate? I understand that doing a hyperparameter search for every experiment is prohibitively expensive, but it could again be done separately in a small-scale setting.
>
> - We clarify that we followed the learning rate settings of previous works (Mix-LN and LayerNorm-Scaling) strictly and didn't specifically tune learning rate for our approach. We believe that directly using their optimal learning rate doesn't benefit our approach, but instead shows the good generalization of our approach.
> - Although we did not conduct systematic learning rate search for all model scales and architectures, we did perform a small search for the 350M models. We found that learning rate of 1e-3 tend to cause spikes in early training stages for Pre-LN, and causes the loss to diverge for 3 of the 6 baseline architectures (Post-LN, DeepNorm, Sandwich-LN). Therefore we adopted a learning rate of 5e-4 for this setting.
>
> >Q1. Is there any negative impact of training speed? I would expect not, but it is always better to ask. I think it would be better to clearly mention this in the paper.
>
> Based on our training framework and hardware, the training throughput after adding the GPAS becomes marginal as the model size scales up. The throughput percentage relative to baseline is around 97% for 71M to 350M models, and 99% for 1B to 7B models. Moreover, we believe that implementing the GPAS operation as a fused kernel will further decrease overhead, which our current implementation does not. We will add this observation to our revised version.
>
> >Q2. Since you decided to use a relatively established Commonsense170K dataset/benchmark for evaluation, why not reporting the performance on all datasets in the benchmark?
>
> We chose the subset to align with the results of previous works (Mix-LN and LayerNorm Scaling), since they also reported these subtasks only. Another minor reason is to fit the table within the page width without using a too-small font, which could impact readability. For full disclosure, the complete scores:
>
> (**bold**: higher score between GPAS and base model; ***bold italic***: highest among all)
> |Model|MMLU|BoolQ|PIQA|SIQA|HellaSwag|WinoG|ARC-e|ARC-c|OBQA|AVG|
> |-|-|-|-|-|-|-|-|-|-|-|
> |Post-LN|22.95|37.83|52.88|32.91|26.24|49.49|27.44|19.11|11.60|31.16|
> |DeepNorm|22.95|37.83|52.61|32.91|26.20|49.64|27.23|19.28|11.60|31.14|
> |DeepNorm+GPAS|**26.46**|**53.82**|**67.95**|**33.42**|**31.43**|**50.67**|**47.22**|**22.78**|**21.20**|**39.44**|
> |Pre-LN|25.95|50.37|68.72|**32.91**|32.29|50.99|49.33|21.50|17.80|38.87|
> |Pre+GPAS|**26.68**|**59.76**|**68.99**|32.45|**33.72**|**52.25**|**49.87**|**22.01**|**22.20**|**40.88**|
> |Sandwich-LN|27.29|61.74|67.63|**33.62**|32.70|**51.07**|47.43|23.29|21.20|40.66|
> |Sandwich+GPAS|**27.41**|**61.96**|**69.10**|33.27|**34.62**|50.36|**50.97**|23.29|**22.20**|**41.46**|
> |Mix-LN|**26.24**|61.90|68.82|**33.37**|33.11|***52.88***|48.91|**24.06**|20.80|41.12|
> |Mix+GPAS|26.23|**61.93**|**70.18**|32.96|**33.52**|***52.88***|**49.79**|22.95|**21.00**|**41.27**|
> |LNS|26.62|***62.08***|69.37|***34.34***|34.68|**52.09**|51.18|23.38|20.00|41.53|
> |LNS+GPAS|***27.81***|61.59|***70.95***|32.14|***36.09***|52.01|***52.40***|***25.77***|***24.40***|***42.57***|
>
> >Q3. Is it correct to the SFT evaluation as zero-shot when the models are directly trained on training samples from the evaluated datasets?
>
> The SFT stage is mainly for learning the instruction following capabilities in order to perform the benchmarks properly. We agree that our benchmarks are not ideally zero-shot by using in-domain training data, except for MMLU which does not have a training set. We will clarify that in the updated version.
>
> >Q4. In the experiments with predefined gates, what is the impact of this setting on the activation and gradient variance?
>
> Predefined gates introduces even slightly smaller activation variance and slightly larger gradient, but falls way short in perplexity. Results are shown in the following table:
>
> |Method|Pre-LN|Pre+GPAS|Pre+GPAS (predefined gates)|
> |-|-|-|-|
> |Layerwise Variance|0~120|0~60|0~40|
> |Layerwise Grad Norm|0.05~0.07 | 0.05~0.3|0.05~0.35|
> |Perplexity|21.35|20.35|22.46|
>
> [1] Zhao, J., Zhang, Z., Chen, B., Wang, Z., Anandkumar, A. and Tian, Y., 2024. Galore: Memory-efficient llm training by gradient low-rank projection. ICML 2024.
>
> [2] Li, P., Yin, L. and Liu, S., 2024. Mix-ln: Unleashing the power of deeper layers by combining pre-ln and post-ln. ICLR 2025.
>
> [3] Sun, W., Song, X., Li, P., Yin, L., Zheng, Y. and Liu, S., 2025. The curse of depth in large language models. arXiv preprint arXiv:2502.05795.

---

> > ### Comment · Reviewer_odJN · 2025-08-06
> >
> > Thank you for clearly responding to all me comments.
> >
> > > We believe the MMLU score was caused by using an older version of lm-eval. After updating to the latest version, here are the re-evaluated results:
> >
> > Thank you for the update. The scores still seem to be very close to the 25% random baseline, perhaps an easier benchmark could better highlight the differences between these relatively small models.
> >
> > > Nonetheless, this does not undermine our core claims. On Commonsense Reasoning benchmarks—which are more closely aligned with the C4 domain—GPAS still showed consistent and meaningful improvements.
> >
> > I believe the paper could still greatly benefit from an out-of-distribution evaluation set that tests the generalization properties of your method.
> >
> > > We clarify that we followed the learning rate settings of previous works (Mix-LN and LayerNorm-Scaling) strictly and didn't specifically tune learning rate for our approach. We believe that directly using their optimal learning rate doesn't benefit our approach, but instead shows the good generalization of our approach.
> > > Although we did not conduct systematic learning rate search for all model scales and architectures, we did perform a small search for the 350M models. We found that learning rate of 1e-3 tend to cause spikes in early training stages for Pre-LN, and causes the loss to diverge for 3 of the 6 baseline architectures (Post-LN, DeepNorm, Sandwich-LN). Therefore we adopted a learning rate of 5e-4 for this setting.
> >
> > Thank you for clarifying that, this indeed makes the results stronger.
> >
> > _____
> >
> > I'm quite confident with keeping the score given to this paper the same, I believe that 5 (accept) is a fair assessment for the contributions of this paper.

---

### Decision · Program_Chairs · 2025-09-17

**Decision:**

Accept (poster)

**Comment:**

This paper introduces a method for faster convergence speed. The method is scaling down the intermediate activations while keeping their gradients unchanged. The method is a simple technique that can be used in combination with existing approaches and many transformers variants. The paper is well-written and the experimental results show the effectiveness of GPAS clearly.